# Defining expansions and perturbations to the RNA polymerase III transcriptome and epitranscriptome by modified direct RNA nanopore sequencing

Ruth Verstraten[1,2], Pierina Cetraro[1], Amy H. Fitzpatrick[1,8], Yasmine Alwie[1], Yannick N. Frommeyer[3], Elene Loliashvili[1], Saskia C. Stein [1], Susanne Häussler [3,4,5,6], Werner J. D. Ouwendijk [7] & Daniel P. Depledge [1,2,6] ✉

RNA polymerase III (Pol III) transcribes cytosolic transfer RNAs (tRNAs) and other non-coding RNAs (ncRNAs) essential to cellular function. However, many aspects of Pol III transcription and processing, including RNA modifications, remain poorly understood, mainly due to a lack of available sensitive and systematic methods for their analysis. Here, we present DRAP3R (Direct Read and Analysis of Polymerase III transcribed RNAs), a modified nanopore direct RNA sequencing approach and analysis framework that enables the specific and sensitive capture of pre-mature Pol III transcribed RNAs. Applying DRAP3R to distinct cell types, we identify previously unconfirmed tRNA genes and other novel Pol III transcribed RNAs, thus expanding the known Pol III transcriptome. Critically, DRAP3R also enables discrimination between co- and post-transcriptional RNA modifications such as pseudouridine (Ψ) and $N^6$-methyladenosine (m$^6$A) at single-nucleotide resolution across all examined transcript types and reveals differential Ψ installation patterns across tRNA isodecoders and other ncRNAs. Finally, applying DRAP3R to epithelial cells infected with Herpes Simplex Virus Type 1 reveals an extensive remodelling of both the Pol III transcriptome and epitranscriptome. Our findings thus establish DRAP3R as a powerful tool for systematically studying Pol III transcribed RNAs and their modifications in diverse cellular contexts.

The Pol III transcriptome comprises in excess of 400 tRNA genes[1] and at least 12 gene families that may comprise large numbers of multi-copy genes and pseudogenes (reviewed in ref. 2). The ncRNAs encoded by these genes effect critical roles across diverse cellular processes including translation regulation, splicing, tRNA processing, RNA stability, and immune signalling. Crucially, recent ChIP-Seq based studies have inferred the existence of additional Pol III transcribed genes, most of which are of unknown function[3,4], indicating that the full extent of the Pol III transcriptome has yet to be determined.

3' end processing is crucial for successful gene expression, and this extends to both Pol II and Pol III ncRNAs. Pol III transcription termination initiates with recognition of a short nucleotide (nt) poly(T) tract located within the coding DNA strand[5]. In mammalian genomes, the poly(T) tract is usually 4–6 nt in length[6] and may contain degenerate elements (e.g. TTATTT)[7]. These degenerate elements are

generally associated with tRNA genes and induce low-level disruption of 3' end processing and result in Pol III transcribed RNAs with extended 3' trailing sequences[7,8]. Following transcription termination, the 3' poly(U) tract of Pol III transcribed RNAs is rapidly bound by the N-terminal domain of the La protein[9] to facilitate clearance of newly-terminated RNA from the transcription complex[10–12], trimming of the poly(U) tail, and the recruitment of additional proteins involved in RNA processing[9].

The role of RNA modifications in the biology of tRNAs and other Pol III transcribed RNAs remains an area of intense study. While the addition of modifications to such ncRNAs is often described as a post-transcriptional process[2], there remains ample evidence that many such modifications are installed either co-transcriptionally or during the association of pre-mature RNAs with the La protein (reviewed in refs. [13,14]). The list of modifications that have been reported on select La-associating pre-tRNAs include pseudouridine ($\Psi$), N2,N2-dimethyl-guanosine ($m^2_2G$), $N^1$-methyladenosine ($m^1A$), and $m^5$-cytosine ($m^5C$)[15–19]. The extent to which pre-mature Pol III RNAs are modified prior to or during binding of La protein remains unknown as a systematic examination of the pre-mature Pol III epitranscriptome has yet to be achieved.

The direct RNA sequencing (DRS) methodology developed by Oxford Nanopore Technologies (ONT) enables the end-to-end sequencing of native RNA molecules[20] and, through subsequent interrogation of signal data obtained during sequencing, permits the identification of specific chemically modified nucleotides, including $m^6A$ and $\Psi$, within the sequence itself[21–25]. While primarily designed for the selective capture and profiling of poly(A) RNAs, alternative strategies have been developed and successfully applied to the capture and sequencing of (i) individual RNAs[23], (ii) mature tRNAs[26,27], and (iii) the global transcriptome[28]. While the latter was achieved by size-separation and polyadenylation of all RNAs present in a population, the targeted sequencing of mature tRNAs and, on an individual basis, other non-adenylated RNAs, was achieved through modification of the existing ONT adaptors that are ligated onto RNA targets.

Reasoning that a similar approach would enable of a full systematic survey of the Pol III transcriptome and epitranscriptome, we developed the Direct Read and Analysis of Polymerase III transcribed RNAs sequencing (DRAP3R) methodology. In contrast to the above methodologies, we apply a subtractive approach to deplete non-poly(U) RNAs prior to ligation with a custom poly(U)-targeting adapter designed for the latest RNA004 chemistry, thus enabling the capture of pre-mature poly(U)-tailed RNAs. We highlight specific computational challenges associated with basecalling, adapter trimming, alignment, and the robust identification of $m^6A$ and $\Psi$ modifications within individual reads. We demonstrate our ability to sensitively and reproducibly obtain high yields of reads that span, and expand, the known Pol III transcriptome across multiple cell types including primary cells. We further demonstrate that discrete patterns of site-specific installation of pseudouridine is a common feature of pre-mature pre-tRNA and that perturbations to the Pol III transcriptome and epitranscriptome may be induced by external stressors such as viral infection.

## Results

### Subtractive selection and RNAse A treatment enables efficient capture of poly(U) RNAs

To enable capture of poly(U) RNAs, we developed DRAP3R which uses a custom nanopore adaptor to bind 3' poly(U) tracts of four nucleotides or longer. Our custom adaptor replaces the 10 nt poly(T) overhang of standard nanopore DRS adapters and replaces it with a 6 nt overhang sequence of 3' NNAAAA 5' (Supplementary Fig. S1a), thus enabling the capture of any RNA with a poly(U) tract of 4 nt or longer. The standard DRS approach relies on the isolation of poly(A)$^+$ RNA from total RNA prior to adaptor ligation, reverse transcription, motor

protein ligation, and sequencing[20], a selective approach that is designed to minimize overloading of the flowcell. In the absence of a robust poly(U) enrichment approach, we opted for a subtractive approach in which we deplete poly(A)+ RNAs and ribosomal RNAs (rRNAs) prior to ligation with a custom adaptor (Fig. 1a). We stabilize adaptor-ligated RNAs by reverse transcription and subsequently introduce RNAse A to specifically degrade unadapted ssRNAs (circRNAs, mature tRNAs, nascent RNAs, miRNAs, etc.) prior to a final cleanup and motor protein ligation. Resulting libraries are then loaded onto RNA specific (RA) flowcells for sequencing (Fig. 1a). To validate the DRAP3R approach, we generated and sequenced libraries from six samples representing three distinct cell lines: ARPE-19, a spontaneously arising retinal pigment epithelia cell line; NHDF, normal human dermal fibroblasts; and CRO-AP5, a B cell lymphoma cell line containing latent Epstein Barr Virus (EBV) and Kaposi's sarcoma associated Herpesvirus (KSHV) that expresses high-levels of the well-characterized Pol III transcribed EBER2 RNA.

Each library was sequenced for 24 hrs each using RA flowcells on a MinION Mk.1b and yielded between 0.925–2.611 million reads (Fig. 1b) that, following basecalling without trimming by Dorado v0.7.0, revealed a modal read length (inclusive of the ~70 nt adaptor sequence) of ~140 nucleotides for the ARPE-19 and NHDF samples (Fig. 1b). For the CRO-AP5 dataset the modal value was ~220 nt which corresponds to the length of the highly abundant Pol III transcribed EBER2 RNA[29]. We supplemented these data with an additional run using the DRAP3R adaptor to sequence a pool of in vitro transcribed (IVT) RNAs generated from RN7SK, three tRNAs (tRNA-Arg-ACG-1-1, tRNA-Glu-TTC-2-1, tRNA-Glu-CTC-1-1) and EBV EBER2 templates (Supplementary Fig. S1b) and a standard DRS run on the poly(A) fraction retained from one of the ARPE-19 samples (Fig. 1b, Supplementary Fig. S1c).

### Optimized processing strategies required for poly(A) adapter removal and read alignment

As previously reported[27], minimap2[30] is suboptimal for aligning shorter read lengths (< 200 nt) and a parameter sweep using minimap2 and BWA[31–33] enabled us to identify the optimal alignment parameters (bwa mem −W 13 −k 6 −T 20 −x ont2d) and increase the proportion of aligned reads from 33.2% (minimap2 −ax map-ont) to > 99.9% (Supplementary Fig. S1d). A comparison of read length vs. alignment length for each alignment strategy confirmed that the optimal bwa mem parameters generally aligned the full length of each read, minus the ~70 nt adaptor sequences (Supplementary Fig. S1e), particularly when subsequently filtering to retain only primary alignments (Supplementary Fig. S1f).

Standard DRS datasets benefit from the removal (trimming) of the 3' adaptor sequence prior to alignment, a procedure typically performed during basecalling with Dorado. While highly effective for standard poly(A) datasets, we observed a significant reduction in the ability of Dorado to remove adaptors from our poly(U) datasets (Supplementary Fig. S2a) with more than half of the reads remaining untrimmed and a further fraction being under-, or over-trimmed (Supplementary Fig. S2b). We attribute this to an inability of the Dorado algorithm to effectively identify and segregate the poly(U) signal. While the adaptor sequence is a fixed length, RNA basecalling remains inaccurate, particularly where DNA is passing through the pore, as evidenced by the variable degree of trimming (30-150 nt, modal value ~ 65 nt) on both poly(U) and poly(A) datasets (Supplementary Fig. S2b). This variability precludes the use of a fixed distance trim strategy as a proportion of reads would still be over- or under-trimmed. There is thus an unmet need for improved adaptor trimming strategies when using non-poly(A) adaptors. For this study, we omitted the trimming step entirely, observing that the majority of adaptor sequences were reported as soft-clipped sequences during BWA alignment and thus excluded from downstream analyses.

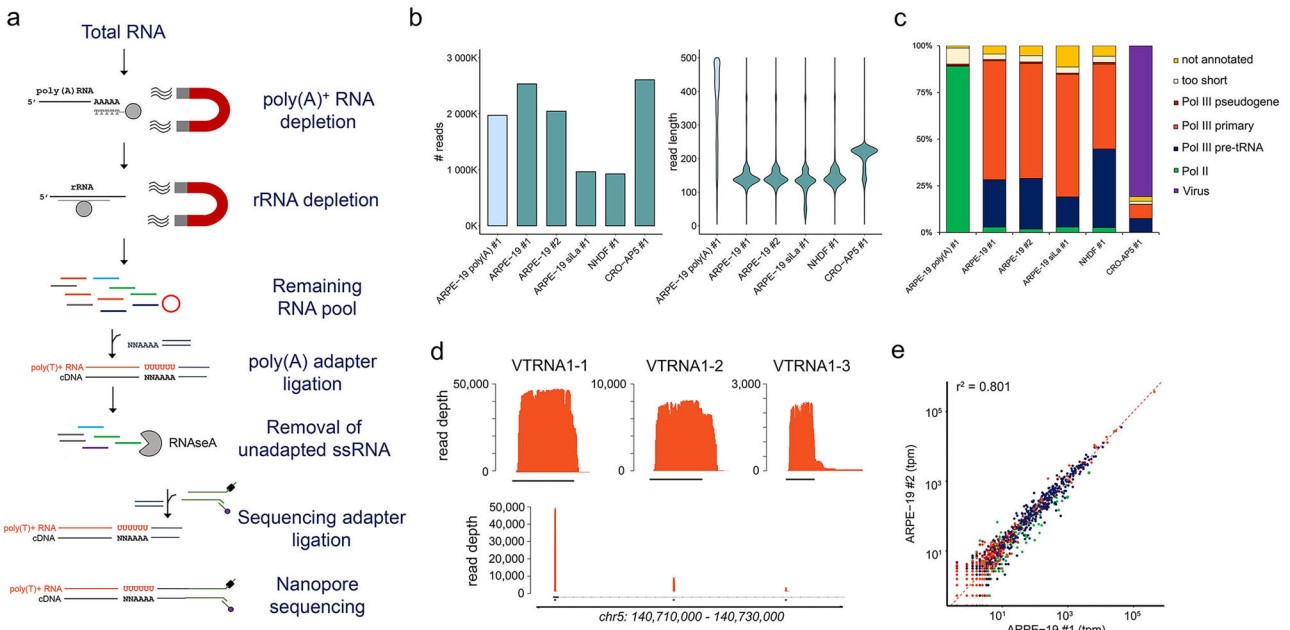

**Fig. 1 | DRAP3R efficiently captures poly(U) RNAs. a** DRAP3R utilizes a subtractive approach to remove poly(A) and ribosomal RNAs in a stepwise fashion, prior to ligating a double-stranded DNA adapter to all RNAs with a 3′ poly(U) tract. Following cDNA synthesis to generate hybrid RNA:cDNA molecules, all remaining single-stranded RNAs are removed by RNAseA digestion. Resulting libraries are loaded onto RNA (RA) flowcells and sequenced on a MinION mk1B for 24 h. **b** Total read count and read length distribution for four DRAP3R runs (2x ARPE-19 biological replicates,1x ARPE-19 pre-treated with a La protein siRNA, Normal Human Dermal Fibroblasts (NHDF), and CRO-AP5 (B cell lymphoma cell line)) and a standard poly(A) DRS run (ARPE-19). **c** Sequence reads were classified as Pol II transcribed (green) or Pol III transcribed with the latter subdivided into pre-tRNAs (dark blue), primary Pol III genes (orange), and pseudogenes (red), the latter defined by their annotation in Gencode v45. Short reads that could not be unambiguously assigned to a specific gene are shown in beige while reads that did not overlap with any existing annotation are shown in gold. Viral reads, derived from EBV present in CRO-AP5 cells are shown in purple (**d**) Read coverage plots across the VTRNA1 locus on chromosome 5 highlight the specificity and full transcript coverage provided by DRAP3R. **e** Scatter plot comparing the abundances of RNAs within distinct classes of RNA Pol II and RNA Pol III transcripts shows very high correlation between biological replicates of ARPE-19 cells. Colour coding as in (**c**).

## DRAP3R is highly selective for poly(U) RNAs

To evaluate the specificity of DRAP3R, we performed intersect analyses between the genome alignments and existing annotations for RNA Pol II and RNA Pol III transcripts (Gencode v45[34]) and tRNA genes (Genomic tRNA Database, GtRNADB[1] (Fig. 1c and Supplementary Table S1). We required alignments to overlap with at least 25% of an annotation to be classified and counted. Approximately 60–70% of read alignments could be assigned to known RNA Pol III transcribed genes (and pseudogenes) including *RNY5, 7SK, RMRP, RPPH1, RNA5S*, and vault RNAs (Supplementary Fig. S3a). The triple repeat of vault RNA genes (*VTRNA1-1, VTRNA1-2*, and *VTRNA1-3*) located on chromosome 5 are well characterized in terms of expression with *VTRNA1-1* the most highly expressed and *VTRNA1-2* and *VTRNA1-3* expressed at increasingly lower levels[35], an observation mirrored in our data (Fig. 1d). As expected, given the generally short lengths of Pol III transcribed RNAs, most read alignments extend to within 5-10 nt of the defined 5′ end of transcripts (Supplementary Fig. S3b). Notably, DRAP3R also enables capture of RNAs with degenerate poly(U) sequences such as *VTRNA1-3* (UCUU) and RN7SL2 (UUUGU).

In general, only a small fraction (0.8-1%) of reads aligned to Pol III pseudogenes. For tRNA genes, GtRNADB predicts 619 tRNA genes within the human genome, 429 of which are reported with high confidence. Restricting our analysis to alignments with mapping quality (mapQ) scores ≥ 10 and tRNA genes with at least five reads aligned, we observe expression of 362 distinct tRNA genes with read alignment counts between 5 to >30,000 (Supplementary Fig. S3c). Of these, 322 tRNA genes were previously classified as high confidence. Of the 40 low-confidence tRNA genes, 24 were supported by 10 or more unique read alignments with six of these supported by 100 s to 1000 s of read alignments, thus providing strong evidence that these are authentic transcription targets (representative examples shown in

Supplementary Fig. S3d). The presence of untrimmed and incorrectly aligned adapter sequences provides a visual indication that the 3′ end of these tRNA genes are not accurately recorded in GtRNADB (Supplementary Fig. S3d). We further observed the presence of T -> C mutations at several positions in these tRNAs. Such mutations are often artefacts derived from the presence of pseudouridine which can cause errors during standard basecalling[25].

Of the remaining reads, ~2–3% could be assigned the defined RNA pol II transcripts (Supplementary Fig. S3e), principally the U1 and U2 genes that, similar to RNA Pol III genes, are characterized by short poly(U) tracts at 3′ end[36,37]. Of the remaining Pol II transcripts captured by DRAP3R, we observed a number of known targets for non-templated polyuridylation[38] including H3C13 mRNAs encoding histones (Supplementary Table S2)[39].

1–3% of all alignments overlapped with existing Pol III and Pol II annotations but were excluded from further analysis for being too short (< 25% length of exonic region). Finally, ~5% of aligned reads could not be assigned to any existing tRNA or Gencode annotations and thus represent potentially novel RNA Pol III transcribed RNAs. Similar distributions were observed for all DRAP3R datasets except for CRO-AP5 which was dominated by the EBER2 RNA (Fig. 1c). Importantly, very high correlation scores were observed for RNA abundances between biological replicates (Fig. 1e) indicating the robustness of DRAP3R.

## DRAP3R expands the RNA polymerase III transcriptome

Recent studies reported the identification of eight putative Pol III transcribed genes based on ChIP-Seq binding profiles of select Pol III subunits[3,4], four of which have since been classified as high-confidence tRNA genes[1] and two as putative tRNA genes. Of the remaining two putative genes, one comprises an Alu/SINE element (AluJb_SINE_Alu)

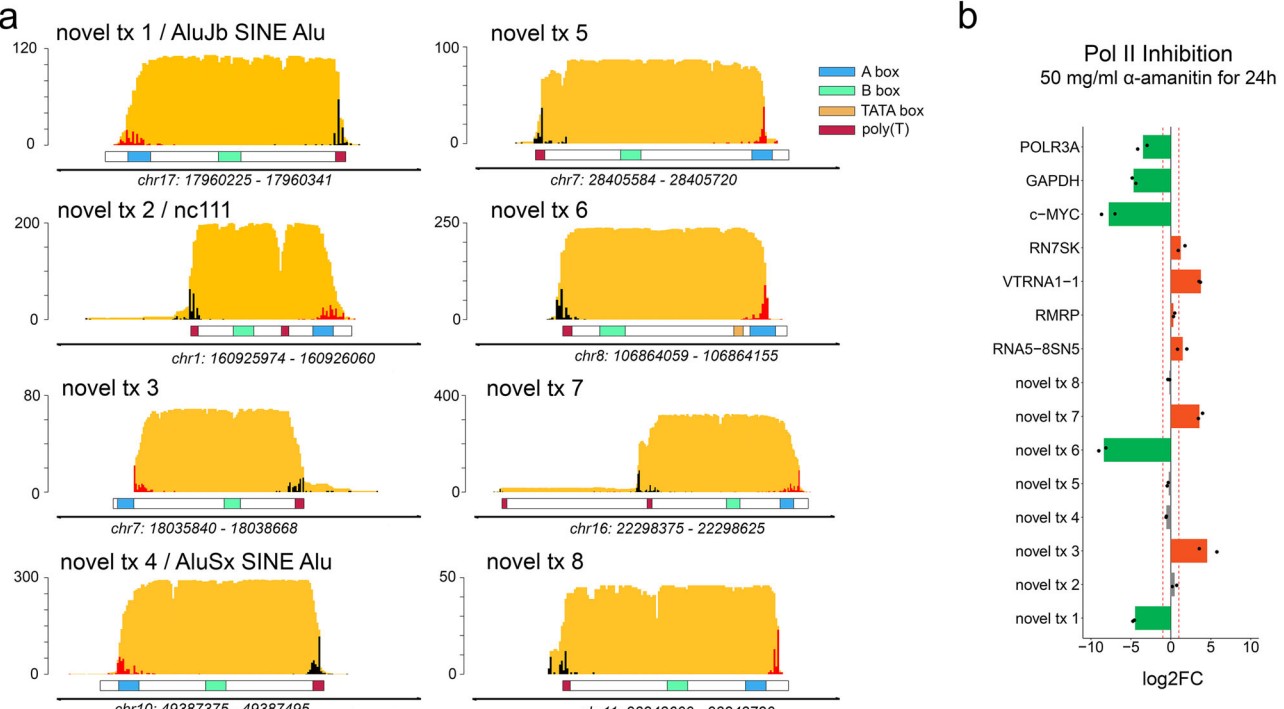

**Fig. 2 | DRAP3R expands the RNA polymerase III transcriptome. a** Coverage plots for eight putative novel RNA Pol III transcribed RNAs are shown in gold with 5′ (red) and 3′ (black) alignment ends shown as overlaid histograms. Characteristic sequence features including A (blue) and B (green) boxes as well as TATA boxes (pink) and poly(T) tracts ≥ 4 nt (purple) are shown as part of the underlying transcript schematic. Genome co-ordinates are specified for the HG38 assembly while the y-axis denotes read depth. **b** ARPE-19 cultures (*n* = 2 biological replicates) were treated with 50 μg/ml α-amanitin for 24 h to selectively inhibit Pol II transcription and quantitative RT-PCR performed on extracted RNA to determine expression levels relative to 18 s rRNA (expressed as log2 fold difference). Selective inhibition of Pol II transcription was confirmed for all Pol II transcribed targets (POL3RA, GAPDH, C-MYC, and RNU2-1).

reported as a Pol III binding target by Canella et al., while the other was named as nc111 by Rajendra et al. Here, DRAP3R identified RNAs originating from each of these locations for all three cell lines, confirming the putative tRNA genes and novel genes as active sites of transcription (Fig. 2, Supplementary Fig. S4a, Supplementary Table 3), Sequence analysis confirmed the presence of sequence elements that show high similarity to A and B box motifs, along with short 3′ poly(T) tracts between 100 and 250 nt downstream of the relevant Pol III subunit binding site (Fig. 2, Supplementary Fig. S4a). This combination of sequence elements is characteristic of type 2 promoter architecture[40,41].

Further examination of our own datasets subsequently identified six additional novel putative Pol III genes, again expressed in all three cell lines, with similar sequence elements (Fig. 2 and Supplementary Table 3). Sequence homology analyses identified no matches to known RNA species. Given that a subset of Pol II transcribed RNAs (e.g. U1 and U2) also terminate at short poly(U) tracts, we treated cultured ARPE-19 cells for 24 h with differing concentrations (50–250 μg/ml) of α-amanitin, an inhibitor of Pol II and Pol III transcription, the latter in a dose-dependent manner. We subsequently assayed expression levels of the 8 candidate Pol III RNAs alongside four defined Pol III RNAs and four defined Pol II RNAs relative to the Pol I transcribed 18 s rRNA. At 50 μg/ml we observe a specific reduction in expression of all Pol II transcribed RNAs along with two of the candidate Pol III RNAs (novel tx 1 and 6) while all Pol III transcribed RNA levels, including those of the remaining six novel RNAs, were unchanged (Fig. 2b). These observations remained consistent at higher α-amanitin concentrations (150 and 250 μg/ml) although the latter showed, as expected, a relative reduction in Pol III transcripts levels, consistent with increasing levels of α-amanitin induced transcription inhibition (Supplementary Fig. S4b, c). Together, these data demonstrate that the Pol III transcriptome comprises additional uncharacterized RNAs, the expression of which is conserved across all three cell types.

## Silencing of La protein selectively destabilizes pre-tRNAs

A key step in the protection of pre-mature RNAs transcribed by Pol III is the binding of La protein which acts as a chaperone to prevent degradation by 3′ exonucleases[42]. To determine whether specific subsets of Pol III RNA are particularly sensitive to La protein levels, we utilized a DRAP3R ARPE-19 dataset (Fig. 1b, c) in which La protein expression was reduced by pre-treatment with an siRNA for 72 h prior to RNA harvest (Supplementary Fig. 1g). Here, pre-tRNA level were reduced relative to untreated ARPE-19 datasets (Fig. 1c) suggesting that pre-tRNAs may be more sensitive to La protein levels than other Pol III derived RNAs. To test this, we calculated decay rates (difference in coverage between 5′ and 3′ ends) for all Pol III derived RNA isoforms and compared these values for Pol III genes and Pol III tRNA genes between two untreated biological replicates and the La protein knockdown dataset (Fig. 3). While no differences in decay rates were observed for Pol III genes between the different datasets (Fig. 3a, b), a number of tRNA genes showed evidence of increased decay rates when La protein was knocked down (Fig. 3c, d, and Supplementary Data 1). These data were verified by comparing coverage plots for select non-affected RNAs (RMRP and tRNA-Asn-GTT-7-1, Fig. 3e, f) and affected RNAs (tRNA-Trp-CCA-2-1 and tRNA-Tyr-GTA-2-1, Fig. 3g, h), albeit with the caveat that additional biological replicates would increase the stringency of these analyses and should provide a focus for future studies.

## Pseudouridine modifications are characteristic of pre-mature RNA Pol III transcripts

Sequencing RNAs in their native state enables the detection of select RNA modifications. The Dorado basecaller (v0.7.0) released by ONT

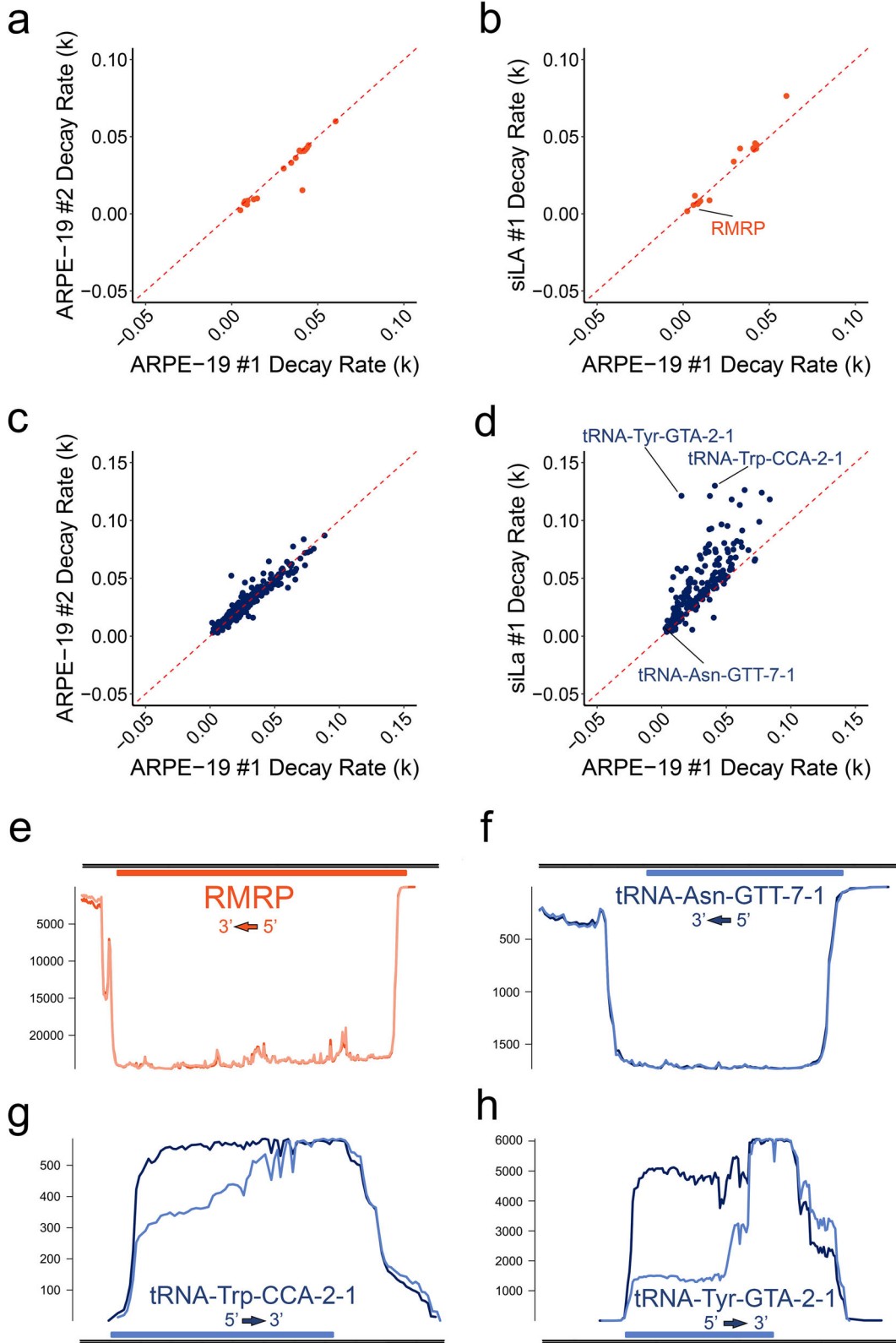

**Fig. 3 | Silencing of La protein selectively destabilizes pre-tRNAs. a–d** 5′ - > 3′ decay rates (*k*) were calculated for all RNA derived from Pol III genes and tRNA genes in each of the ARPE-19 datasets. Scatter plots comparing decay rates for Pol III derived (**a**) pre-mature RNAs (orange) and (**c**) pre-tRNAs (dark blue) in untreated ARPE-19 cells were highly correlated, as were Pol III derived pre-mature RNAs when comparing between untreated and siLa treated ARPE-19 cells. By contrast, (**d**) Pol III derived pre-tRNAs showed increased decay rates in siLA treated ARPE-19 cells. **e–h** Strand-specific coverage plots for a representative Pol III derived pre-mature

RNA (RMRP) and three pre-tRNAs. The Y-axis denotes depth of coverage while the underlying gene model is shown in orange (non-silencing control) and light orange (siLa) for RMRP or dark blue (non-silencing control) and light blue (siLa) for the tRNA genes. Representative RNAs with consistent decay rates include (**e**) RMRP, and (**f**) tRNA-Asn-GTT-7-1, while RNAs showing decreased stability when La protein is depleted include (**g**) tRNA-Trp-CCA-2-1 and (**h**) tRNA-Tyr-GTA-2-1. Directionality is indicated on each panel.

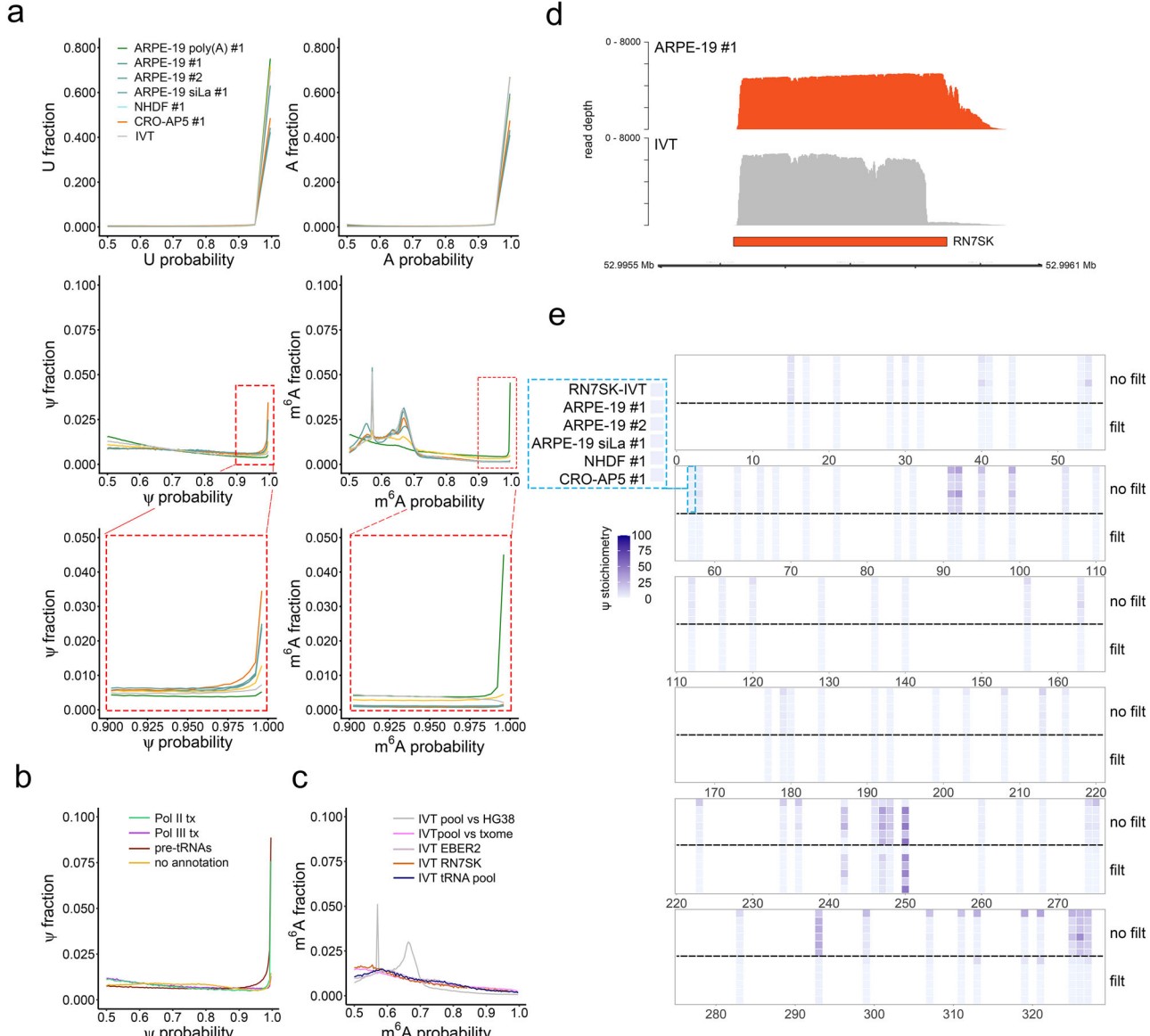

**Fig. 4 | Detection of Ψ and m⁶A and in pre-mature Pol III transcribed RNAs.**
**a** Basecall probability distributions for U, A, Ψ, and m⁶A in all datasets generated in this study were calculated using modkit. Datasets and colours are indicated in the upper left panel. **b** Basecall probability distributions for Ψ, and m⁶A in dataset ARPE-19 #1, separated according to annotation category. Pol II RNAs - green, Pol III RNAs - purple, Pol III pre-tRNAs – dark red, not annotated–gold. **c** m⁶A basecall probability distributions for the pooled IVT dataset following alignment against the HG38 genome (grey) or transcriptome (pink). Distributions derived from analysis of IVT EBER2 (plum), RN7SK (orange), and the pooled IVT tRNAs (dark blue) are also shown. **d** Coverage plots of pre-mature RN7SK RNA in the DRAP3R ARPE-19 #1 and RN7SK IVT datasets. Y-axis denotes read depth while X-axis denotes co-ordinates on HG38 chromosome 6. The RN7SK gene annotation is shown as a grey box. **e** Ψ stochiometry distributions on pre-mature RN7SK demonstrates a high-false positive rate in unfiltered datasets (top five rows) but not in filtered datasets (bottom five rows) which show pseudouridine installation at positions 243, 247, and 250 (blue asterisks).

---

supports native detection of (all-context) $N^6$-methyladenosine (m⁶A) and pseudouridine (Ψ) and assigns a confidence score (probability) to each individual canonical and modified nucleotide basecall. This enabled us to test the hypothesis that Ψ installation on Pol III transcribed RNAs occurs prior to post-transcriptional processing of the poly(U) tail. Following super accuracy (sup) basecalling, we plotted the basecall probability distributions for A vs. m⁶A and U vs. Ψ for each dataset and observed near-identical distributions for canonical A and U nucleotides across all datasets (Fig. 4a and Supplementary Fig. S5a). High confidence (modification probability > 0.98) Ψ basecalls accounted for 0.5–1% of all U sites and were enriched in DRAP3R datasets relative to standard poly(A) DRS and IVT datasets, the latter of which do not contain any modified ribonucleotides (Fig. 4a and Supplementary Fig. S5b). We next generated modification probability

distributions after first sub-setting read alignments based on their annotation as canonical Pol III genes, canonical Pol III tRNA genes, and Pol II genes. A fourth category represented reads that did not align against any existing annotations, including the putative Pol III derived RNAs reported in this study. Here we observed that high confidence Ψ basecalls are enriched in pre-tRNAs and Pol II-derived poly(U)-tailed RNAs (Fig. 4b). High confidence m⁶A basecalls accounted for 0.5–1% of all A sites in the standard poly(A) DRS dataset but were significantly reduced in the DRAP3R (and IVT) datasets (Fig. 4a and Supplementary Fig. S5b). Two unexpected peaks, with modification probability values of ~0.57 and ~0.68) were observed in the m⁶A modification probability distributions derived from both biological and pooled DRAP3R datasets. Interestingly, these appear when deriving modification probability distributions from genome-aligned datasets but not when

aligning IVT data against a simplified transcriptome containing the five individual transcript sequences, suggesting an alignment artefact (Fig. 4c).

The presence of numerous false-positive Ψ and m⁶A basecalls in our IVT datasets (Supplementary Fig. S5b) was indicative that an additional filtering strategy would be required to reduce false-positives while preserving true-positives. A close inspection of the modification probability distribution plots confirmed that Ψ and m⁶A basecalls with confidence scores > 0.98 were relatively rare in the IVT datasets compared to the main datasets. We thus reasoned that filtering to retain only high-confidence basecalls (confidence score > 0.98) would effectively eliminate false-positive modification calls. To examine the effect of this filter, we compared the location and stoichiometry of Ψ and m⁶A in pre-mature RN7SK RNAs. We observed similar coverage profiles between DRAP3R and IVT datasets (Fig. 4d) but discordant patterns in Ψ predictions (Fig. 4e). Here, many putative Ψ sites are consistently predicted in the unfiltered data but are absent from our filtered dataset containing high confidence Ψ calls. As these sites were also reported in the pseudouridine-free IVT datasets, we characterized these as false positives. Notably, the well-characterized Ψ250 in RN7SK[43] was observed in all filtered DRAP3R datasets and absent in the IVT dataset. Additional, lower stoichiometry positions were observed at Ψ243 and Ψ247, also in agreement with a prior observation[44]. These observations thus provide confidence in the specificity and sensitivity of Ψ detection following our filtering strategy.

An analysis of putative m⁶A sites within RN7SK showed an absence of any installation at this pre-mature stage (Supplementary Fig. S6a), consistent with previous reports that m⁶A is post-transcriptionally installed on RN7SK RNA[23]. A similar analysis was conducted on the EBV EBER2 RNA for which Ψ114 and Ψ160 have been discordantly reported[45,46]. Here, we obtained good coverage of EBER2 in DRAP3R and IVT datasets (Supplementary Fig. S6b) and detected Ψ at position 114 but not 160 (Supplementary Fig. S6c). We observed no evidence of m⁶A installation on the pre-mature RNA (Supplementary Fig. S6d).

We next examined the distribution of Ψ and m⁶A stoichiometries across all filtered datasets (Supplementary Fig. S7a), segregated into two distinct groups, Pol III transcribed pre-tRNAs and all other Pol III transcribed RNAs. For both Ψ and m⁶A, a majority of putatively modified positions had stoichiometries below 10%. For Ψ, ~32% of Ψ installation sites had stoichiometries above 10% while for m⁶A this value was reduced to ~5%. Notably, larger numbers of Ψ and m⁶A sites were predicted in pre-tRNAs relative to other pre-mature Pol III RNAs. These observations remained consistent across all datasets, leading to the conclusion that high stoichiometry Ψ, but not m⁶A, is a feature of pre-mature Pol III transcribed RNAs.

Metaplots visualize the density of a given modification across the body of RNAs. To validate the accuracy of the high-confidence Ψ and m⁶A basecalling, we observed that, consistent with previous studies[46–48], m⁶A and Ψ distributions across mRNAs in our poly(A) DRS dataset showed characteristic enrichment of m⁶A at the boundary of the CDS and 3′ UTR and a general enrichment of Ψ across the CDS (Supplementary Fig. S7b). For the DRAP3R datasets, we observed that m⁶A and Ψ installation is enriched at specific regions in pre-tRNAs and more generally toward the 5′ end of other pre-mature Pol III transcripts (Supplementary Fig. 7c) and, in the case of Ψ, regardless of whether filtering for only high stoichiometry positions ( > 10%) (Supplementary Fig. S7d).

## DRAP3R enables identification of Ψ and m⁶A on pre-mature small nuclear RNAs

While some prior reports have identified individual pre-tRNAs with modified ribonucleotides[15–19], the extent to which the pre-mature Pol III transcriptome is modified prior to or during binding of La protein has yet to be systematically evaluated. To address this, we examined the

installation profiles for m⁶A and Ψ across the pre-mature Pol III transcribed RNAs captured by DRAP3R. We restricted our analysis to sites at which m⁶A/ Ψ stoichiometry exceeded 10% in at least one dataset because prior nanopore and targeted-quantification studies recover 10% synthetic mixtures reliably and BID-seq/BACS-style approaches treat sites above ~10% as confident[46,49,50]. Excluding pre-tRNAs, we detected 11 m⁶A sites (Supplementary Fig. S8a) and 24 Ψ sites (Fig. 5a) across fifteen transcript isoforms. The methyltransferase(s) responsible for m⁶A installation on pre-mature Pol III transcribed RNAs is not known, and no sequence motif could be derived from these sites (Supplementary Fig. S8b). We detected three (Ψ31, Ψ40, and Ψ86) of four Ψ modifications previously reported for U6 RNA[51–55] and two sites in U6atac (Ψ9, Ψ83), the latter of which was also previously reported[56] (Fig. 5a). Interestingly, while Ψ211 is reported for RN7SL RNAs[46,57], we observed no evidence of Ψ in these RNAs in any of our datasets. Here, we additionally report the discovery of Ψ sites located in RNY4, RNY5, RMRP, BC200, RNA5S, and the conserved 5′ sequences of VTRNA1-1, VTRNA1-2, and VTRNA1-3 (Fig. 5a). No motif could be established for the Ψ sites (Supplementary Fig. S8c), many of which are known to be installed by distinct pseudouridine synthases (reviewed in ref. 58). Notably, we observed a complete absence of Ψ sites on the pre-mature Pol II transcribed U1 and U2 RNAs that are also captured by DRAP3R, indicating that Ψ installation on pre-mature poly(U) RNAs is generally restricted to pre-tRNAs and select Pol III genes.

## Pseudouridine installation is a characteristic marker of pre-tRNAs

An analysis of our DRAP3R pooled IVT dataset demonstrated that, similarly to RN7SK and EBER2, unfiltered pseudouridine basecalls produced very high false positive detection rates that could be eliminated when filtering to retain only modification probabilities ≥ 0.98 (Supplementary Fig. 8e). We subsequently generated a quantitative map of Ψ installation across all pre-tRNAs in our ARPE-19 datasets (Fig. 5b). Ψ sites at high stoichiometry were consistently detected at positions 13, 27–28, 31/32, 35/36, 55, and 65 with positions 55 and 13 having the highest stoichiometry across all datasets, including those from NHDF and CRO-AP5 cells (Supplementary Fig. 9). Discrete combinations of Ψ installation were observed for different pre-tRNAs, both within and between isotypes, as shown for glutamic acid tRNA genes (Fig. 5c). These data are in agreement with a recent study of Ψ sites across the human transcriptome[46] while also providing greater specificity by identifying discrete patterns at the level of individual tRNA genes and showing that significant pseudouridylation of pre-tRNAs occurs during transcription and/or La binding (Fig. 5d). This ability of DRAP3R to discriminate between pre-tRNAs at the gene level provides a significant advantage when compared to methods that require collapsing to the level of isodecoder or isotype[27,46]. Based on the presence vs. absence of Ψ at positions 13 and 55, we ascertained consensus motifs for pre-tRNA pseudouridylation by Pus7 (UVΨAR) and Pus4 (VNΨCR) (Fig. 5e), respectively, the former matching that of a previous report[46].

Having established that pre-tRNAs carry pseudouridine modifications, we next asked whether other known modifications of tRNAs are evidenced at the pre-mature pre-tRNA level or are only manifest as part of post-transcriptional processing. While modifications such as 1-methylguanosine (m¹G) are not yet available for native identification during basecalling, prior studies have demonstrated that RNA modifications are frequently associated with increased basecalling errors around the site of the modification[26,27]. To enable comparisons between pre-tRNA and mature tRNAs we adapted the nano-tRNA-Seq[27] method for the latest DRS RNA004 chemistry, sequenced mature tRNAs from the same cultured ARPE-19 cells and observed near-identical Ψ installation profiles across glutamic acid isodecoders (Fig. 5f). We then examined differences in basecall error rates for DRAP3R and nano-tRNA-Seq datasets at each individual position

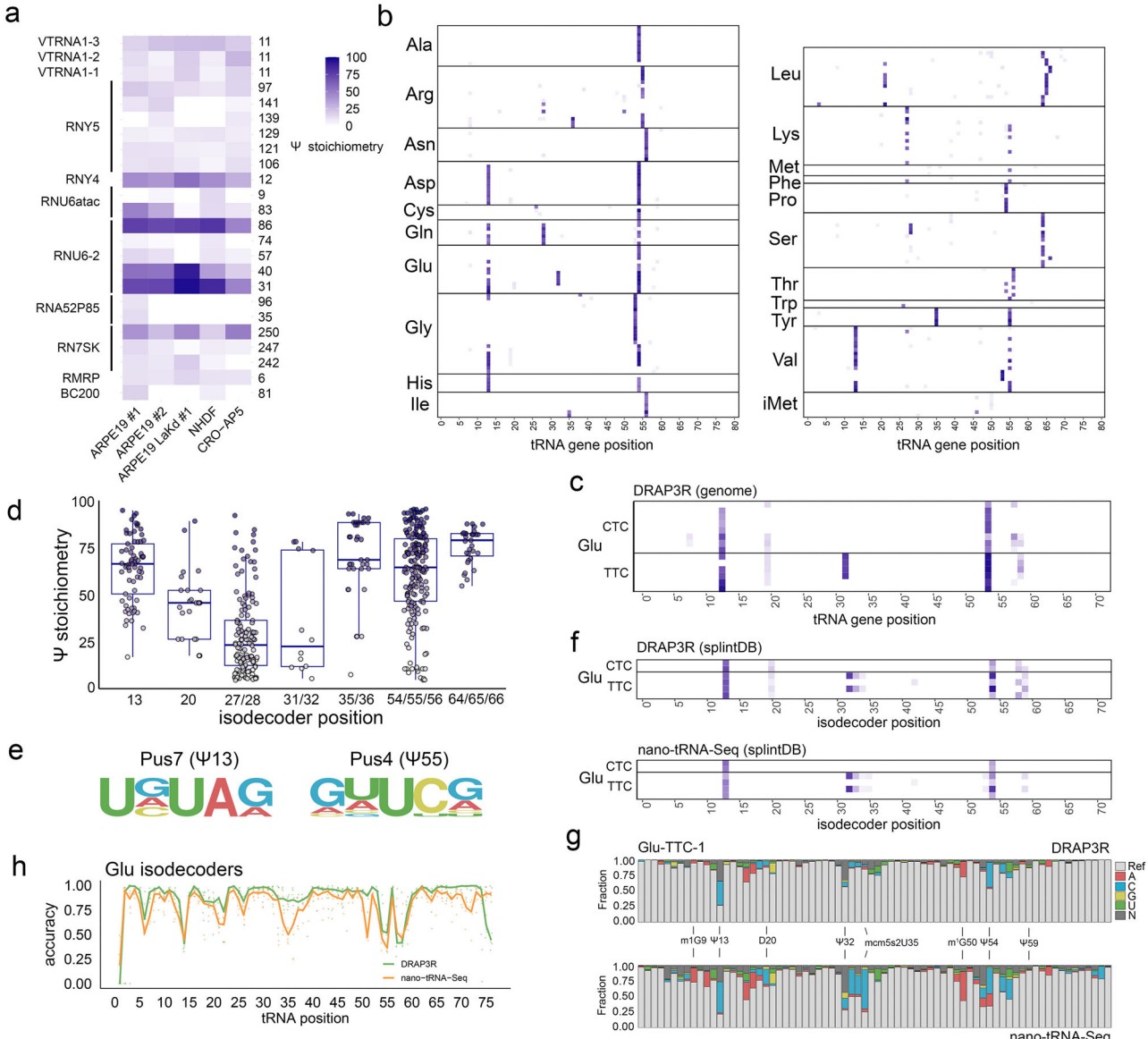

**Fig. 5 | Ψ profiles across the pre-mature Pol III transcriptome. a** Heatmap showing stoichiometry distributions > 10% across Ψ sites in pre-mature Pol III transcribed ncRNAs vary between cell types. **b** Ψ profiles across all pre-mature pre-tRNAs with coverage ≥ 50 reads in the DRAP3R ARPE-19 #1 dataset. Each row depicts different tRNA genes and rows are grouped by isoacceptor. **c** As (**b**) but showing only detected glutamic acid tRNAs genes. **d** Box and whisker plots (plots (center = median, box = 25th–75th percentiles, whiskers = values within 1.5 × IQR, outliers plotted individually) denoting Ψ stoichiometry distributions ( > 5%) at

defined Ψ positions across all pre-tRNAs for the ARPE-19 #1 dataset. **e** Consensus sequences motifs for Ψ installation by Pus7 at Ψ 13 and Pus4 at Ψ 55. **f** As (**c**) except that the ARPE-19 #1 DRAP3R and nano-tRNA-Seq datasets were aligned against a tRNA isodecoder splint database. **g** Distribution of basecall errors for the ARPE-19 #1 DRAP3R and nano-tRNA-Seq datasets across the Glu-TTC-1 isodecoder. Select modified nucleotides are labelled. **h** Basecall error rate distributions for the ARPE-19 #1 DRAP3R and nano-tRNA-Seq datasets across each position of all glutamic acid isodecoders.

across the glutamic acid tRNA-TTC-1 gene (Fig. 5g) and observed notable differences in error rates at positions 9 (m¹G), 20 (dihydrouridine, D), 35 (5-methoxycarbonylmethyl-2-thiouridine, mcm⁵s²U) and 50 (m¹G), indicating that these positions are post-transcriptionally modified. By contrast, error rates at positions 13, 32, 55, and 59 (all Ψ) were similar in both datasets, confirming this modification is installed co-transcriptionally or while bound by La. Analysis of all glutamic acid isodecoders yielded similar overall results (Fig. 5h), while also demonstrating isodecoder specific differences e.g. between Glu-TTC-2 and Glu-TTC-4 (Supplementary Fig. S8d). This demonstrates the ability of DRAP3R, in combination with nano-tRNA-Seq, to distinguish between pre-mature and post-transcriptional tRNA modifications, even where basecalling models are lacking the ability to natively detect specific modifications.

## DRAP3R reveals infection with HSV-1 increases the relative abundance of pre-tRNAs while reducing pseudouridine stoichiometries

We and others have previously demonstrated the ability of HSV-1 to regulate m⁶A installation on mRNAs[59,60] and that HSV-1 infection leads to changes in pre-tRNA and mature tRNA levels[61], but no study has addressed whether pseudouridine installation is also regulated. To address this, we applied DRAP3R to RNA collected from two biological replicates each of HSV-1 infected ARPE-19s collected at 6 h post infection (hpi) and 12 hpi. We obtained similar numbers of reads and read length distributions to the other DRAP3R datasets (Figs. 1b, 6a) while classification showed a slight increase in the relative proportions of pre-tRNA and other Pol III transcripts captured (Figs. 1c, 6b). The biological replicates were significantly

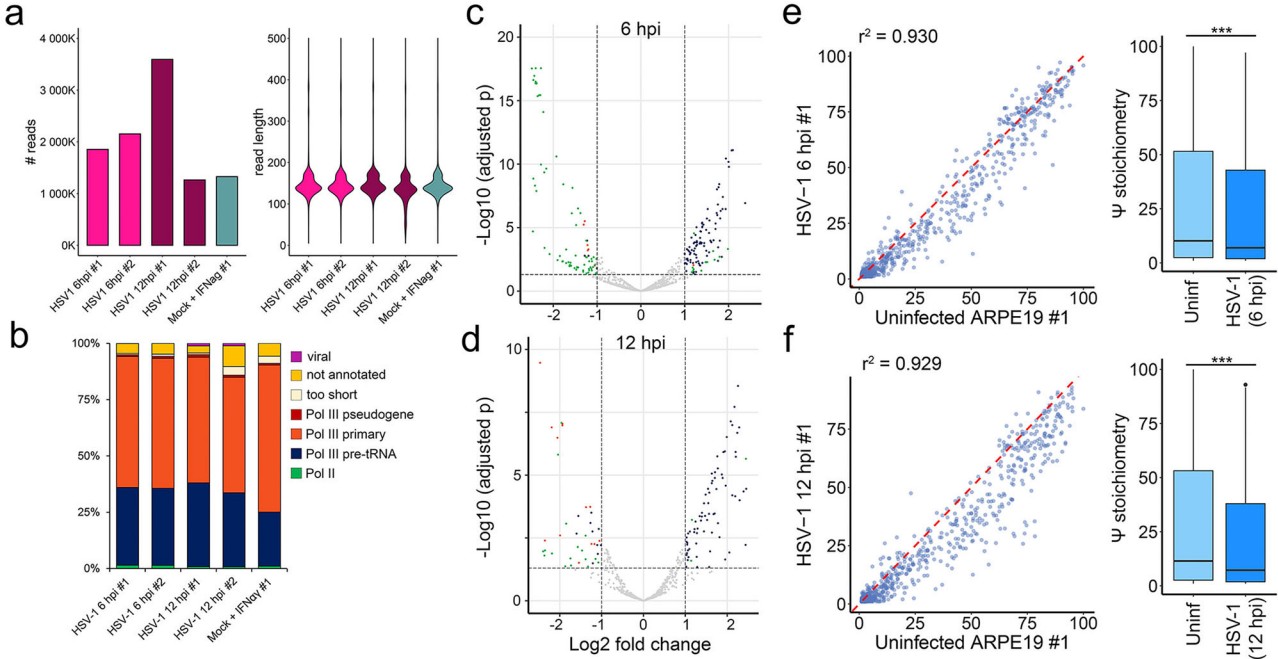

**Fig. 6 | HSV-1 infection regulates pre-tRNA expression and Ψ installation.**
**a** Total read count and read length distribution per DRAP3R run using RA flowcells with RNA004 chemistry and a minimum read length cutoff of 20 nt. **b** Sequence reads were classified as Pol II transcribed (green) or Pol III transcribed with the latter subdivided into pre-tRNAs (dark blue), primary Pol III genes (orange), and pseudogenes (red), the latter defined by their annotation in Gencode v45. Short reads that could not be unambiguously assigned to a specific gene are shown in beige while reads that did not overlap with any existing annotation are shown in gold. Viral reads, derived from EBV present in CRO-AP5 cells are shown in purple (**c**–**d**) Volcano plots showing significantly ($p$adj < 0.05) differentially regulated pre-tRNAs (dark blue), Pol II RNAs (green) and Pol III genes (orange) at (**c**) 6 hpi and (**d**) 12 hpi

relative to mock-infected cells ($n$ = 2 biological replicates per condition / timepoint). Differential transcript expression was determined using DeSeq2 with shrunken LFC obtained using the adaptive shrinkage estimator (*ashr*) and adjusted p values derived using the Benjamini–Hochberg (BH) procedure. **e**–**f** Scatter plot comparing Ψ stoichiometries on pre-tRNAs shows decreasing correlation (Spearman's $r^2$) between uninfected and HSV-1 infected (**e**–6 hpi, **f**–12 hpi) ARPE-19 cells ($n$ = 2 biological replicates per condition / timepoint). Underlying data are additionally represented as bar and whisker plots (center = median, box = 25th–75th percentiles, whiskers = values within 1.5 × IQR, outliers plotted individually) with p values calculated using Student's paired $T$ test (two-sided).

correlated (Supplementary Fig. 10a) while basecall accuracy probability distributions for A vs. m⁶A and U vs. Ψ (Supplementary Fig. 10c) and Ψ stoichiometry distributions (Supplementary Fig. 10d) were also consistent with the previous DRAP3R runs (Supplementary Fig. 7a). Differential gene expression analyses showed, in agreement with a previous study[61], a significant upregulation in expression of ~50 pre-tRNAs and significant downregulation of most Pol II transcribed genes at both 6 and 12 hpi (Fig. 6c, d and Supplementary Data 2, 3). We further observed strong correlations in site-specific Ψ frequencies between both uninfected (Supplementary Fig. 10e) and infected (Supplementary Fig. 10f, g) biological replicates. Notably, comparisons of site-specific Ψ stoichiometries between uninfected and HSV-1 infected ARPE-19s revealed decreasing Ψ stoichiometries that correlate with the length of the infection (Fig. 6e, f) and that this is not a result of interferon induction (Supplementary Fig. 10h). With the caveat of limited timepoints ($n$ = 2) and bioreps ($n$ = 2 per condition), this observation provides a basis for further studies of HSV-1 mediated regulation of pseudouridylation during infection. Finally, a prior ChIP-Seq[61] study indicated that Pol III can bind the HSV-1 genome and therefore may be capable of transcribing directly from the viral genome. Here, we observed small numbers of HSV-1 poly(U) RNAs (Supplementary Fig. 11) aligning proximal to previously defined Pol II transcription start sites[62,63]. While this may point to Pol III transcription of the viral genome, it is important to note that (i) a feature of many alphaherpesviral genomes is the presence of numerous poly(T) tracts in both coding and intergenic regions[64], and (ii) Pol II transcription from viral genome is extensive such that at any given timepoint, numerous nascent RNAs are in the process of being

transcribed. Our DRAP3R methodology may thus simply be capturing Pol II derived RNAs mid-transcription at poly(T) tracts.

## Discussion

Pioneering studies in the 1970s/1980s provided the first evidence that pre-mature pre-tRNAs originating from Pol III transcription are modified either co-transcriptionally or while bound to protein La as a first step in post-transcriptional processing[15–19]. The role of such modifications in orchestrating the first steps of processing remain poorly characterized, limited by the lack of a systematic method for examining RNA modifications on individual pre-mature RNAs. This lack is driven by the challenges in capturing small RNAs for analysis and the difficulty in distinguishing between such RNAs at different stages of post-transcriptional maturation. A direct consequence of this is that only a minority of human tRNAs, and far fewer pre-tRNAs, have been characterized in terms of their RNA modifications[65]. Further, efforts to generate an atlas of the full Pol III transcriptome has been hindered by the general reliance on ChIP-Seq based studies to putatively identify previously unreported Pol III transcribed RNAs[3,4]. This similarly impacts the ability to screen for changes induced in the Pol III transcriptome and epitranscriptome by internal and external stressors.

To address these limitations, we developed DRAP3R, the first nanopore DRS based method that specifically and sensitively captures RNAs with short 3' poly(U) tails, a unifying characteristic of all known RNAs transcribed by Pol III. In contrast to the limited number of existing nanopore DRS-based methods for profiling ncRNAs, which generally rely on adding poly(A) tails in vitro prior to adapter ligation, we have devised a custom adaptor that specifically targets poly(U) tails and combined this with a subtractive approach to remove non-poly(U)

tailed RNAs. In combination with native basecalling and stringent filtering of Ψ and m⁶A modifications, we demonstrate our ability to delineate and measure changes in the discreet patterns of modifications installed on pre-mature ncRNAs, thus enabling a clear discrimination of modifications installed co-transcriptionally or during La-binding from those which are added during subsequent post-transcriptional processing steps. We further identify multiple novel RNAs that are generally insensitive to low concentrations of α-amanitin treatment, indicating they are transcribed by Pol III. The two identified RNAs that were sensitive to α-amanitin also require further study to deduce whether these are transcribed by Pol II or simply require Pol II activity for Pol III transcription to occur, as has been described for U6 RNAs[66].

DRAP3R is simple to implement with the entire protocol, including sequencing, requiring 2–3 days. The high data yields obtained, combined with the promise of recently proposed DRS multiplexing approaches[67,68], further increase the potential of our method. We should, however, note current limitations that still need resolving. Firstly, a recurrent issue observed is that adapter trimming during basecalling of DRAP3R datasets with Dorado frequently leads to over-trimming while retaining the adapter sequence can lead to misalignments that extend beyond the 3' end of sequenced RNAs. There is thus an unmet need for improved adapter identification and removal approaches that are not reliant on the presence of poly(A) tails. Secondly, while short poly(U) tails serve as markers of pre-mature RNAs transcribed by Pol III and are rapidly removed upon La binding, the multi-copy U6 RNA exhibits a unique processing pathway. Here, pre-mature U6 RNAs undergo polyuridylation to generate an extended poly(U) tail, which is subsequently trimmed during post-transcriptional processing, ultimately yielding mature U6 RNAs with poly(U) tails ranging from 5 to 12 nucleotides in length[69]. It is thus likely that a proportion of U6 RNAs captured by the DRAP3R protocol are not purely representative of pre-mature RNAs. Similarly, we also show that DRAP3R is capable of capturing polyuridylated RNAs such as HIST2H3D/H3C13 and pre-let-7, indicating that our method could be adapted to facility studies of alternative RNA decay pathways. Thirdly, while count data derived from DRAP3R allows for relative quantification analyses, the absence of a spike-in IVT RNA remains a limitation. Finally, as with other DRS-based protocols, DRAP3R does not provide sequence information for the extreme 5' end of RNA molecules.

The recent development of other nanopore-based methods for targeted analysis of mature tRNAs[26,27] or the entire cellular transcriptome[28] offer significant potential in combination with DRAP3R, allowing for the comparison of relative abundances and the modification status of pre-mature and fully mature and charged versions of tRNAs and other ncRNAs transcribed by Pol III. This is demonstrated by our integration of DRAP3R and nano-tRNA-Seq which reveals that pseudouridine is generally installed co-transcriptionally or during La binding while modifications such as m¹G are added at a later post-transcriptional stage. This finding highlights the value of comparative sequencing of pre-mature and mature Pol III transcribed RNAs in determining the timing of modification installation and we anticipate the value of this approach will only increase in the future as nanopore basecallers are updated to include native detection of additional RNA modifications. As a point of caution however, we observed that modification predictions generated by the Dorado basecaller produce significant numbers of false-positives, as evidenced by m⁶A and Ψ modification calls across multiple IVT datasets. Our filtering strategy was sufficient to remove the vast majority of false-positives without comprising detection of true-positives reported in other studies using orthologous approaches. However, the thresholding approach applied does not establish a null model, nor control false-positive rates. Care is thus required when interpreting results using this approach and highlights the unmet need for robust ground truth datasets for establishing efficient and reproducible approaches to filtering Dorado-based modification detection calls.

Another potential application of DRAP3R is in the study of Pol III-mediated Pol II transcription. Rajendra et al.[3] recently reported on the transcription of short-lived poly(U)-tailed Pol III transcribed RNAs occurring from a subset of Pol II promoters as a necessary step for Pol II transcription initiation. While such RNAs were not observed in our datasets, presumably due to their instability and rapid degradation, modified experimental designs that prevent rapid turnover of such RNAs could provide sufficient targets for DRAP3R. Similarly, where internal (i.e. inborn errors) or external stressors (i.e. viral infection) may exert impacts on Pol III transcriptome and RNA modification installation, the combination of DRAP3R and nano-tRNA-Seq would allow researchers to establish at which co- and post-transcriptional steps the presence of specific modifications are required.

In summary, DRAP3R is a fast, sensitive, and accurate method for quantitative profiling of the RNA Pol III transcriptome and the detection of RNA modifications at single-nucleotide resolution on individual RNAs. We anticipate a wide range of applications for this method in broadening our understanding of Pol III transcription and modification dynamics, including the role of the Pol III transcriptome and epitranscriptome in infection biology and the study of human disease induced by inborn errors in ncRNAs and Pol III subunits.

## Methods

### Cell culture, RNA extraction, and virus preparation
ARPE-19 cells were grown in DMEM/F12 with 8% FCS + 1% P/S + 1% L-glutamine. Cells at 70% confluency were treated with 50, 150 or 250 µg/ml of α-amanitin for 24 h. Cells were harvested by adding Trizol and RNA extracted according to manufacturer's protocol with addition of GlycoBlue (Thermo-Fisher) during the first precipitation step. cDNA was synthesized using the Quantitect reverse transcription kit (Qiagen), followed by q-PCR using the QuantiNova SYBR green (Qiagen) and primers (Supplementary Table 4) using 18sRNA to normalize. The qPCR programme consisted of 40 cycles of 10 s at 95 °C and 30 s at 60 °C and a melting curve at the end to check for product specificity. For HSV-1 infections, ARPE-19 cells were grown to 90% confluency and cell-free HSV-1 strain KOS added at an MOI of 10 in 2% FCS media and incubated for 1 h at 37 °C with 5% $CO_2$. After 1 h the infection media was removed and fresh media with 8% FCS was added and cells were harvested at 6 or 12 hpi. ARPE-19 cells grown to 70% confluency were treated with interferon-α2a and interferon-γ (from ImmunoTools) 500 and 100 U, respectively, for 24 h before harvesting. CRO-AP5 cells were obtained from the DSMZ (ACC 215) and grown in RPMI 1640 (Gibco) with 20% FCS (Sigma).

### La protein silencing
ARPE-19 cells were grown to ~70% confluency in a 6-well plate. An siRNA duplex (25 pmol, Sigma-Aldrich) targeting La protein was transfected using RNAiMax (Invitrogen). The siRNA duplex sequence was previously published in Rajendra et al., 2024[3]. Total RNA was harvested in Trizol and protein collected using RIPA buffer, both at 72 h post-transfection. For immunoblotting, 20 µg of total protein from cells lysed in Ripa buffer was loaded on a 4–12% Bis-Tris gel (Sigma-Aldrich: MP41G10) and run with MES buffer. Protein was subsequently transferred onto a PVDF membrane and blocked for 2 h with 5% milk in PBS-tween, La/SSB Antibody (312B) (Santa-Cruz:sc-80656) was used at a 1:100 concentration and incubated overnight at 4 °C.

### DRAP3R
To remove poly-adenylated RNAs from 20–25 µg of total RNA, we used the Dynabeads™ mRNA Purification Kit (Invitrogen) and retained the supernatant after the first pulldown of the magnetic beads as the non-poly(A) fraction and re-isolated the RNA by ethanol precipitation. Following poly(A) depletion, 10 µg of remaining RNA was used for

rRNA depletion using the RiboMinus ™ Transcriptome Isolation Kit (Invitrogen), also followed by an ethanol precipitation. The concentration of the resulting poly(A)/rRNA⁻ RNA was determined by Qubit. We used 500 ng of this RNA for Direct RNA Sequencing library preparation using the RNA-SQK004 chemistry (Oxford Nanopore Technologies), with some modifications to the standard protocol. Here, we replaced the RTA adapter with our custom DRAP3R adapter (Supplementary Fig. 1a). The DRAP3R adapter replaces the 10 nt poly(T) sequence with 3' NNAAAA 5' for specific capture of poly(T) RNAs. To prepare the adapter, we combined 1.4 μM each of RNAse-free HPLC purified Oligo A and DRAP3R Oligo B in 100 μl of buffer (10 mM Tris-HCL pH 7.5, 50 mM NaCL), heated this to 95 °C for 2 min followed by cooling (0.1 °C/sec) to 4 °C. During the first adapter ligation we also omitted the polyadenylated RNA calibration strand (RCS). Following reverse transcription with SuperScript III (Invitrogen), RNAse A (43 μg/ml final concentration) in the presence of 0.4 M NaCl was added and incubated at 37 °C for 15 min to remove all remaining single-stranded RNA. RNAse A was subsequently inactivated by proteinase K treatment for 15 min at 37 °C, followed by 70 °C for 15 min. Ligated of the RLA adapter was performed according to the ONT protocol with all purification steps using AMPure XP beads at 2X concentration. The resulting sequencing library was loaded onto an RNA-specific (RA) minION flowcell and sequencing performed for 24 h with MinKNOW parameters modified to disable q-score filtering and reduce the minimum read length for retention to 20 nt.

### In vitro transcription of RN7SK and EBER

To produce templates from which in vitro transcribed (IVT) RNAs could be generated, we used genomic DNA extracted from MeWo or CRO-AP5 cells on which a PCR was performed using Q5-high-fidelity polymerase and custom primers where the forward primer contains the T7 promoter (Supplementary Table 4). This was then followed by transcription using the TranscriptAid T7 High Yield Transcription Kit (Thermo-Fisher) following the manufacturers instruction for high yield. Note that only canonical dNTPs (A, C, G, and U) were provided. The resulting RNA was isolated using the Monarch RNA clean up kit 50 μg (NEB) and yielded ~100 μg of IVT RNA. 100 ng of each IVT reaction was added (RN7SK, EBER2, tRNA-Arg-ACG-1-1, tRNA-Glu-TTC-2-1, and tRNA-Glu-CTC-1-1) to a total of 500 ng was used as input for the DRAP3R DRS library preparation, omitting the poly(A) and rRNA depletion steps.

### Nanopore sequencing of poly(A) RNA

The poly(A) fraction was isolated from 25 μg of total input RNA using the Dynabeads™ mRNA Purification Kit (Invitrogen), according to manufacturer's protocol. 300 ng of poly(A) RNA was used as input for the standard nanopore DRS RNA004 protocol with 45 ng loaded onto an RNA specific RA flowcell prior to sequencing on a MinION Mk.1b. for 24 h.

### Genomes and databases

The human genome (GRCh38.p14) was obtained via Ensembl[70] while the human transcriptome (v45) was obtained via GENCODE[71]. The human tRNA database (GRCh38 Dec 2013) was downloaded from GtRNADB[1]. The HSV-1 strain KOS genome (KT899744[72],) and EBV strain B95-8 genome (AJ507799.2[73],) were downloaded from Genbank. Hybrid genome indexes (e.g. HG38 + HSV-1 strain KOS) were generated by merging fasta files and indexing using BWA [*bwa index -a bwtsw hybrid.fasta hybrid.fasta*] or minimap2 [*default parameters*][30].

### Nanopore basecalling

Basecalling was performed using Dorado v0.7.0 (Oxford Nanopore Technologies) in super-accuracy mode with modification detection enabled for Ψ and m⁶A. Each dataset was processed with and without adapter trimming with the latter used for all major downstream analyses. Resulting unaligned BAM files were parsed using SAMtools[74] to generate fastq files with modification tags for each read included in the fastq read headers [*samtools fastq -TMM,ML*].

### Genome and transcriptome level alignments

To define an optimal alignment strategy fastq reads were aligned against the HG38 genome assembly using distinct strategies including BWA SW v0.7.17[33] [*bwa bwasw, bwa bwasw -z 10 -a2 -b1 -q2 -r1*], BWA MEM v0.7.17[31] [*bwa mem -W 13 -k 6 -x ont2d, bwa mem -W 13 -k 6 -T 20 -x ont2d, bwa mem -W 13 -k 6 -T 10 -x ont2d, bwa mem -W 9 -k 5 -T 10 -x ont2d*], and Minimap2 v2.26[30] [*minimap2 -ax map-ont, minimap2 -ax splice -k15 -uf -L*]. Prior to alignment, the fastq dataset was filtered to remove all reads longer than 1,000 nt and divided into bins comprising 250k reads. Resulting SAM files were processed (view, merge, sort, index) using SAMtools v1.18 with only primary alignments [-F 2308] retained. For all downstream analysis, the optimal genome alignment strategy was defined as *bwa mem -W 13 -k 6 -T 20 -x ont2d*.

For alignment of standard nanopore DRS datasets again the human transcriptome we used minimap2 [*-ax map-ont -y -L -p 0.99*]. SAM files were subsequently processed (view, merge, sort, index) using SAMtools[74] with only primary alignments [-F 2324].

### Processing of RNA modification calls using Modkit

Sample probability distributions were obtained using sample-probs subcommand from Modkit v0.4.1 (https://github.com/nanoporetech/modkit) with the following parameters [*--hist --only-mapped --percentiles 0.1,0.2,0.3,0.4,0.5,0.6,0.7,0.8,0.85,0.9,0.95 --num-reads 250000*]. Note that the –num-reads value was reduced to 20000 for each IVT dataset. Relevant columns were extracted from the probabilities.tsv output files and plotted. To generate extended bedMethyl files for which base modifications counts from every sequencing across each reference genomic position are tabulated, we used the Modkit pileup subcommand [*modkit pileup in.bam out.bed --filter-threshold 0.8 --mod-thresholds a:0.98 --motif A 0 --ref HG38.fasta*]. Here, we retain canonical nucleotide calls (ACGU) with >80% confidence scores while retaining only Ψ and m⁶A calls (17802 and a, respectively) with >98% confidence scores, as informed by our analysis of sample probability distributions (Fig. 4a and Supplementary Fig. 5). We further restricted modification call analysis to positions matching the canonical nucleotide in the reference genome (e.g. A for m⁶A and T for Ψ).

### Statistical analysis of modification profiles

To assess differences in modification probability between experimental conditions and the IVT MIX control, we computed per-bin differences across the modification probability range. For each bin, the difference between the condition and IVT MIX was calculated and transformed into a z-score relative to the mean and standard deviation of all bin-wise differences. Bins with an absolute z-score ($|z|$) ≥ 2 were classified as significantly different from the control.

### Intersect analyses and generation of count data

BEDtools v2.26.1[75] intersect [*-s −F 0.25 −wao -a alignment.bed −b specific.database.bed*] was used to classify read alignments in bed and bedMethyl datasets as deriving from transcription of Pol III primary genes, Pol III tRNA genes, Pol III pseudogenes, or Pol II genes. Read alignments with no overlap against a combined Pol II & Pol III gene database were identified using exclusion parameters [-s −v -a alignment.bed −b combined.database.bed]. Novel Pol III transcription was only considered where 50 or more read alignments against the same unannotated locus were reported. Reads aligning to HSV-1 were identified by filtering for the HSV-1 strain KOS ID in the first (chromosome) column of the alignment.bed files. Count data for each gene was collected by summing the total number of primary alignments against each gene.

## Quantification of RNA stability using an exponential decay model

To estimate the relative stability of Pol III transcribed RNA, we modelled the 5′ to 3′ coverage decay using a single-parameter exponential decay function. Nanopore DRS alignments intersected with annotated Pol III gene features (inc. tRNAs) were generated using bedtools intersect. Reads mapping to the same gene locus were grouped and coverage aggregated along each transcript length. Read depth across the aligned region was binned into 5 nt windows for each gene and the mean coverage per bin calculated and plotted as a function of distance from the 5′ end of the transcript. Coverage values were then fit to a nonlinear exponential decay model:

$$C(x) = A \cdot e^{\{-kx\}}$$

where $C(x)$ is the coverage at position $x$, $A$ is the initial coverage at the 3′ end, and $k$ is the decay constant. Fitting was performed using non-linear least squares (scipy.optimize.curve_fit in Python), with $k$ interpreted as a proxy for tRNA turnover rate i.e. larger $k$ values indicate faster decay and lower stability. Only genes with a minimum number of mapped reads (e.g. ≥100) and sufficient coverage across bins were retained for fitting. The resulting decay constants were compared between conditions to assess relative changes in stability.

## Analysis and visualization of modification stoichiometries

Stoichiometry distribution histograms, metaplots, transcript-level plots, and scatter plots were generated using Rstudio/Posit and R v4.3.3 with the following libraries: data.table, patchwork, scales, tidyr, Dplyr, readr, tools, gdata, stringr, Biostrings, EnhancedVolcano, DESeq2[76], GenomicFeatures[77], ggplot2[78], Gviz[79]. Access to scripts used for analysis and visualization is detailed below.

## Reporting summary

Further information on research design is available in the Nature Portfolio Reporting Summary linked to this article.

## Data availability

Raw pod5 datasets generated as part of this study are available via the ENA under the accession number PRJEB81484. R scripts used in the analysis and visualization of data presented herein are available from the https://github.com/DepledgeLab/DRAP3R repository while underlying data are included in the Source Data files and Zenodo repositories (https://doi.org/10.5281/zenodo.17853128, https://doi.org/10.5281/zenodo.17854176, https://doi.org/10.5281/zenodo.17876201, (https://doi.org/10.5281/zenodo.17865866), and https://doi.org/10.5281/zenodo.17856257). Source data are provided with this paper.

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

## Acknowledgements

D.P.D. is supported by a German Centre for Infection Research (DZIF) Associate Professorship and the NIAID grants R01-AI170583 and R01-AI152543. E.L. is supported by the NIAID grant R01-AI170583. D.P.D. and S.H. receive funding from the Deutsche Forschungsgemeinschaft (DFG, German Research Foundation) under Germany's Excellence Strategy - EXC 2155 - project number 390874280." P.C., E.L. and Y.F. were supported by the Hannover Biomedical Research School (HBRS) and the Center for Infection Biology (ZIB). W.J.D.O. is supported by the National Institute of Allergy and Infectious Diseases (NIAID) of the National Institutes of Health (NIH) under Award Number R01-AI151290. The funders had no role in study design, data collection and analysis, decision to publish, or preparation of the manuscript. The authors would additionally like to thank Thomas Hennig and Lars Dölken for useful discussions about this project.

## Author contributions

D.P.D. and R.V. conceptualized and designed the study. R.V. and P.C. performed the experiments with additional support from Y.N.F., E.L. and S.C.S. D.P.D., R.V., P.C., A.H.F. and Y.A. analysed the data. D.P.D., R.V., P.C. and W.J.D.O. interpreted the results. D.P.D. wrote the manuscript with input from R.V., P.C., S.H. and W.J.D.O. All authors commented on the manuscript.

## Funding

## Competing interests

S.C.S. owns a small number of shared in Oxford Nanopore Technologies. All other authors declare no competing interests.

## Additional information

¹Institute of Virology, Hannover Medical School, Hannover, Germany. ²German Center for Infection Research (DZIF), partner site Hannover-Braunschweig, Hannover, Germany. ³Institute for Molecular Bacteriology, TWINCORE GmbH, Center of Clinical and Experimental Infection Research, a joint venture of the Hannover Medical School and the Helmholtz Center for Infection Research, Hannover, Germany. ⁴Department of Molecular Bacteriology, Helmholtz Center for Infection Research, Braunschweig, Germany. ⁵Department of Clinical Microbiology, Copenhagen University Hospital—Rigshospitalet, Copenhagen, Denmark. ⁶Cluster of Excellence RESIST (EXC 2155), Hannover Medical School, Hannover, Germany. ⁷Department of Viroscience, Erasmus Medical Center, Rotterdam, The Netherlands. ⁸Present address: School of Veterinary Medicine, University College Dublin, Belfield, Dublin, Ireland. ✉e-mail: depledge.daniel@mh-hannover.de

