## [Transparent Peer Review file · Nature Communications]

Quantitative profiling of RNA modifications in an expanded RNA polymerase III transcriptome

Corresponding Author: Professor Daniel Depledge

Version 0:

Reviewer comments:

Reviewer #1

(Remarks to the Author)

The manuscript "Defining expansions and perturbations to the RNA polymerase III transcriptome and epitranscriptome by modified direct RNA nanopore sequencing" by Verstraten et al. implements a new method, which the authors define as DRAP3R (Direct read and analysis of Polymerase III transcribed RNAs), which essentially depletes non-poly(U) RNAs and ligates poly(U)-targeting adapters, thereby targeting Pol III-derived ncRNAs, which possess 3'-oligo(U) sequences owing to the oligoT termination mechanism of Pol III transcription. The major strengths of the current study are (1) the direct sequencing of Pol III-derived RNA species, which provides some level of investigation regarding the chemical modifications of these oligo(U) RNAs - a major focus of the study - and (2) discovery of novel Pol III-derived RNAs. The authors also use the DRAP3R framework to query changes in tRNA levels and pseudouridine stoichiometry following infection. The implementation of this method and the expansion beyond currently annotated Pol III-transcribed genes are valuable contributions that are certain to be of interest across the Pol III field and beyond. However, there are notable concerns with the chemical modification ("epitranscriptome") aspect of the current study that should be addressed, primarily with respect to the sparsity of in vitro transcribed, non-modified RNA species, which appear to be a critical requirement for interpreting all epitranscriptome results.

For example, Figure 2C is our first introduction to inferred pseudouridine stoichiometry of in vitro transcribed 7SK RNA – it was surprising to see such a high level of pseudouridine stoichiometry for non-endogenous RNA, can the authors explain this? While the manuscript notes that filtering reduces false positive rates, it suggests that in vitro transcribed RNA is needed for each and every RNA species in order to interpret modification state. This is also the case for m6A methylation, shown for 7SK in Supplemental Figure S4a, and for the only other RNA species with IVT - EBER2 (Supplemental Figure S4c,d). In addition to this apparent technical challenge, there is a general lack of statistical analysis, such that it remains difficult to interpret. For example, we are told that filtering of IVT false positives facilitates identification of modification state, but what statistical framework was applied to achieve this result? Naturally, these are critically important for interpreting the broader analyses focused on RNA modification - for example, are in vitro transcribed tRNAs not necessary for interpreting the results of Figure 3b (and supplemental Figure S6)? There are then a number of stoichiometry count and density analyses layered on top of this that are difficult to interpret. For example, on the basis of evidence provided for in vitro transcribed 7SK and EBER1, it is possible that analyses and interpretations have more to do with uridine or adenosine densities rather than any true endogenous modification states.

In the event that the technical challenges noted above cannot be addressed, I would reiterate that direct capture and sequencing of oligo(U) ncRNAs and discovery of novel (potentially) Pol III products is significant, and the authors might consider reworking their study to emphasize these findings more prominently.

Major points:

There is a significant assumption that analyses are tied to nascent RNAs in the absence of any method to specifically isolate or enrich for nascent RNA. Though tRNAs undergo 3' cleavage, other Pol III-derived RNA species are not processed in the same manner, and thus oligo(U) alone is not sufficient to describe an RNA molecule as nascent.

From the opposite perspective, the authors likely do not capture any mature or charged tRNA molecules, and thus all modification aspects related to the full dynamic pool of tRNAs is lost using this approach.

A statistical framework is necessary to interpret “highly confident modification sites”

There is no evidence provided in Figure 4 to support the claim that pre-tRNA levels are increasing while pseudouridine stoichiometries is decreasing upon HSV-1 infection. Projecting colors onto a correlation plot is not sufficient, instead a differential analysis is needed to draw “differential”-related conclusions. Though a volcano plot is provided in Supplemental Figure S7, we are not provided any details of what genes are up- and down-regulated.

Minor points:

Figure 1b: The average read length for PolyA+ RNA seems to be abnormally short in length, can the authors explain?

Figure 1c, the legend color of the virus RNA does not appear to match.

Supplemental S1e is missing a legend.

Figure S2c, pol II-specific inhibitor can be used to verify that the novel transcripts are Pol III transcribed; the authors might also consider using ML-60218 for Pol III-specific inhibition

Figure 3a, Why is RNY4 listed separately from other Y RNAs? It might help to group together and label each YRNA. In addition, specific RNA species are missing from this - perhaps most notable absent is 7SL (one of the most abundant Pol III transcripts). Can the authors comment on why 7SL was not captured (beyond the description in line 286)

Figure 3b,c,f are missing legends (we are to assume it's the same as 3a).

Figure 4a. Given that samples are sequenced on individual minlons, how do the authors account for potential batch effects? This might be of particular concern if the libraries were sequenced at very different times, with potential for differences in minION batch and/or pore integrities. Overall, it is again difficult to “differential conclusions” drawn from single replicate comparisons, which the authors appear to be doing with respect to uninfected and infected samples.

Reviewer #2

(Remarks to the Author)

Reviewer #3

(Remarks to the Author)

Verstraten et al. present DRAP3R, a nanopore direct RNA sequencing (DRS) methodology tailored for analyzing Pol III transcripts, leveraging their characteristic poly(U) tails. The approach involves depletion steps, a custom adapter, and specific analysis strategies to identify transcripts and modifications like Ψ and m6A. The authors validate the method, identify novel transcripts, and apply it to study changes during HSV-1 infection. The data hint at some interesting changes to pol III transcript (relative) abundances and modification profiles upon HSV infection, but this would require the analysis of further biological samples to firm up conclusions.

Major Comments

- Could the authors provide empirical evidence, perhaps via supplementary data, to demonstrate the specific advantages gained from the poly(A) and rRNA depletion steps, particularly given the use of a custom adapter designed for target enrichment? Quantifying the impact of these depletions on the yield and specificity of Pol III transcript capture would strengthen the methodological justification.
- The method's reliance on a 3' poly(U) tract means it may be biased against Pol III transcripts with non-canonical termini (e.g., resulting from imperfect termination). A discussion of this limitation, ideally supported by comparative data (e.g., using polyadenylation-based capture on the same polyA and rRNA depleted RNA), is warranted to assess the comprehensiveness of the captured transcriptome.
- Further justification is required for the chosen alignment parameters. Beyond increasing the percentage of aligned reads (Sup Fig 1c), how was the accuracy of these alignments confirmed, particularly considering the potential for spurious alignments suggested by the increase in secondary/supplementary hits?
- The rationale for selecting the 0.98 confidence threshold for modification calls (Ψ and m6A) needs clearer justification. An analysis demonstrating the sensitivity of key findings (like modification stoichiometry) to this specific cutoff value would strengthen the robustness of the conclusions.
- The presence of intermediate-confidence m6A peaks is an interesting finding that merits further discussion. Do these

signals exhibit unique basecalling error profiles, map to positions of other known modifications, or show enrichment in specific transcript types/classes?

- Precision in terminology regarding modification timing and statements regarding sequencing of nascent transcripts. Is the evidence sufficient to distinguish installation 'during transcription and/or La binding' specifically. Is the protocol not capturing rather transcripts prior to 3' end processing?

- Statements regarding transcript expression changes upon HSV-1 infection require careful phrasing to reflect the relative nature of sequencing abundance data. Terms like 'reduces expression' or 'increases pre-tRNA levels' should be clarified as changes in 'relative abundance' within the sequenced library, acknowledging that absolute levels are not directly measured without appropriate normalization controls (which may be challenging here).

- Conclusions drawn about time-dependent effects or correlations with infection length in the HSV-1 experiments should be presented cautiously, acknowledging the number of replicates and time points analyzed (6 hpi and 12 hpi). Explicitly stating these limitations would provide better context for the claims.

- It may be worth explicitly noting in the limitations that, like standard DRS protocols, DRAP3R does not provide sequence information for the extreme 5' end of the RNA molecules.

- Clarity of the figure legends should be improved throughout. Explicit definitions for all visual elements, particularly color coding (e.g., Sup Fig 1d lines, gene classifications in Fig 1c/e), are needed for unambiguous interpretation.

Minor Comments

- Please reconcile the discrepancy between the adapter sequence described in the text (NNAAAA 3' overhang) and that depicted in Figure 1a (appears as AAAAAA).

- The read classification percentages presented in Figure 1c and discussed in the text (lines 162-189) could be presented more clearly to ensure all categories sum appropriately or unassigned portions are explicitly stated.

- In Sup Fig 1d, please clarify if the observed ~60-70nt difference between read length and alignment length corresponds primarily to the unaligned 3' adapter sequence, as expected.

Overall Conclusion

This manuscript introduces a potentially valuable tool (DRAP3R) for studying the Pol III transcriptome and epitranscriptome using nanopore sequencing. Addressing the points raised above, particularly concerning methodological justifications, potential biases, bioinformatics parameter choices, and careful interpretation of the data, would significantly strengthen the study and increase confidence in its findings.

Reviewer #4

(Remarks to the Author)

Verstraten et al. present a novel direct RNA sequencing method for detection of Pol III-dependent transcripts. The authors' approach is the first of its kind to assess this broad class of RNAs via direct RNA sequencing and uniquely facilitates identification of RNA modifications. The authors use their technique to assess three distinct human cell models in addition to IVT RNAs as a control to develop a tailored bioinformatics pipeline with high confidence dissection of these transcripts. This allows identification of putative new Pol III-dependent RNAs and direct calling of pseudouridine and m6A modifications. Their technique is corroborated by prior instances of RNA modification and provides clarity on discordant reports. Finally, they apply the technique to assess RNA shifts during viral infection and identify changes in RNA abundance and modification. The strength of this paper lies in development and benchmarking of the technique. The paper is well written and thorough in their representation of the data and how methodology was developed and verified.

Major comments:

1. I would ask the authors to be more intentional with how they refer to the RNAs distinguished by their technique. I feel a more appropriate term would be premature rather than nascent. This technique successfully identifies RNAs which contain a polyU-track. For most Pol III-dependent transcripts, the 3' polyU tracks are removed during transcript maturation, however the exact rate at which they occur (as well as order relative to other maturation steps) is not clearly defined in humans. Or in some instances, such as U6, which the authors comment on in the discussion, polyU's are still present in the mature transcript.

2. In line with my prior comments, I feel the biological significance of this manuscript would be bolstered by perturbation of at least one of the core steps of Pol III transcript maturation (e.g. La protein interaction, nucleolytic cleavage) and then assessment via DRAP3R. Alternatively, the authors could apply their technique to different subcellular RNA fractions, as the location for some of these RNA modifications is unknown. Such experiments would significantly broaden the scope of this paper and inform the RNA biology of Pol III transcripts.

3. The authors present their dataset in a number of interesting ways, I would ask that they expand on the following topics:
a. Short RNA reads (line 186-187): The authors remark on a proportion of reads which are quite short. I'm curious what these reads map to (type of transcript, location in transcript, etc) and if the stop location may indicate an RNA modification that their reverse transcriptase could not proceed through. Many other studies of Pol III dependent or other highly structured RNAs use

- a TGIRT-based RT step to aid with processivity. Please expand on these short products, including their parent gene loci, size, and whether they're stop position correlates with previously published RNA modifications.
- b. Pol II RNAs (line 183-185): Polyuridylation is a form of RNA processing by which U's are added to the 3' end of mRNA or ncRNA. This would be something I'd expect the authors could identify in their datasets for Pol II products by comparing the gene sequence to that of the potential 3' overhang on their direct RNA read. This would be another very interesting benefit of their novel technique. Please include an analysis or discussion of whether their technique can detect this 3' RNA processing event.
- c. Viral RNAs (Fig 4b): It's a bit hard to see with the color scheme of the column bar, but it appears the authors identified reads mapping to the virus in their dataset—at least there's a legend box for virus. Please expand on these reads and whether they present as bonafide Pol III-dependent transcripts

Minor comments:

1. We would ask the authors to comment on how quantitative DRAP3R is. It does not appear that they introduced IVT spike-ins at early steps in the protocol, thus I am apprehensive of abundance comparisons in an experimental condition such as infection would robustly changes the RNA landscape. This does not change the major conclusions of this paper, but rather presents as a current study limitation that the authors should comment on.
2. To support the authors findings regarding novel Pol III transcripts (Fig. S2), I would ask that they integrate mapping of published ChIP-Seq datasets for the Pol III and II machinery at these gene loci. This would support conclusions regarding RNA polymerase dependency and suggest a classification (Type 1, 2, 3) for Pol III RNAs.
3. Fig. 1C and 4B. Please add a heatmap to expand the data presented in the column bar graphs regarding the RNAs represented in the Pol III-dependent categories.
4. Fig. 1C legend: Can the authors please define what they classify as "Pol III primary" or "Pol III pseudogene", I'm assuming they refer to 5S rRNA, but the existing reference is ambiguous.
5. Please comment on the potential mechanism by which the virus alters Pol III-dependent RNA modification status. For example, what is known about the relative abundance or localization of the modifying enzymes.

Version 1:

Reviewer comments:

Reviewer #1

(Remarks to the Author)

The strength of the DRAP3R method reported by Verstraten et al. lies in its targeted capture and sequencing of poly(U)-tailed RNAs. Though the revised manuscript submitted by the authors makes some improvement in clarity, my primary concern remains with the interpretation of the epitranscriptome data, particularly for tRNAs – which was not addressed. In short, the authors set an empirical modification probability threshold (≥ 0.98) on the basis of rudimentary observational comparisons of endogenous and IVT 7SK, where no modifications should be expected. At this threshold, many putative modification sites in the endogenous data “disappear” in the IVT data, leaving three modification sites in 7SK that are also documented in the literature. This is presented as a “positive control” supporting the threshold choice, but it is shown only qualitatively, via a heatmap, without statistical or quantitative description of enrichment over IVT.

The “0.98” threshold is not part of a formal statistical framework applied to the data, but rather an empirical filter imposed on top of the Dorado basecaller's internal scoring. Dorado's modification confidence scores are posterior probabilities as estimated by the model for a given base in a given sequence context, and they are inherently model- and context-dependent, not universally calibrated across RNA types. In their rebuttal, the authors describe their filter as “building upon the statistical framework embedded within the basecaller,” but this conflates the basecaller's internal probability model with their own threshold choice. The cutoff of 0.98 is derived from two RNAs (7SK and EBER2) where IVT data are available, based on visual divergence in score distributions, without quantitative estimates of false-positive reduction or true-positive retention. While 0.98 may separate modified from unmodified at a handful of documented sites in 7SK and EBER2, there is no evidence this value is optimal or valid for tRNAs, which differ substantially in sequence, structure, and modification density. The same threshold is then applied to tRNAs without tRNA-specific IVT controls or orthogonal validation. The resulting tRNA profiles and conclusions are therefore based on an untested extrapolation, which was not truly quantitative to begin with.

A further major limitation is the lack of biological replicates for many of the modification analyses. The threshold choice, the distribution comparisons between endogenous and IVT RNAs, the tRNA modification profiles, and the infection versus mock modification differences are largely drawn from single-sample datasets or comparisons. Without replication, it is not possible to distinguish genuine biological differences from run-to-run variation in nanopore signal, capture efficiency, or basecalling. This limitation restricts interpretation of differences in modification stoichiometry between conditions or transcript classes.

Experimentally, the authors have now introduced La knockdown experiments (single replicate) which leads to additional, potentially overdrawn conclusions – in this case about decay rates. However, the authors do not comment on the apparent shift in read length distributions in the La knockdown data, which warrants careful consideration. The knockdown condition appears enriched for shorter reads compared to controls, which could directly influence metrics such as calculated decay rates when these are derived from early-to-late gene read ratios. Without controlling for read length distribution, it is unclear whether the reported changes in decay reflect true biological effects or are instead a consequence of altered fragment size profiles.

We remain convinced that the study is valuable for (1) its methodological advance in purifying and sequencing poly(U)-tailed

RNAs, (2) the discovery of novel Pol III transcribed RNA species, and (3) its potential for predicting candidate modification sites from direct RNA nanopore data. However, several descriptive conclusions remain insufficiently supported by the current data: (1) tRNA modification profiles are presented without tRNA-specific control experiments, (2) conclusions from certain perturbations, such as La knockdown, are based on non-replicated datasets, and (3) potential confounding effects of non-uniform read length distributions are now introduced and not addressed.

Minor comments

1. Replication: The APE-19 siLA and NHDF CRO-AP5 datasets each appear to have only a single replicate.
2. Figure 3d (decay rate): Some decay rate values are shown as less than zero; clarification is needed here and in the methods.
3. Figure 3e–h: The figure legend does not specify the meaning of the different colors or indicate the directionality of the tRNAs.
4. Lines 286–287: Incorrect figure reference.
5. Line 395: Incorrect figure reference.
6. Figure 4e / Lines 325–326: The cutoff used for stoichiometry to designate a modification site is unclear. While a threshold is stated at Line 324, the rationale should be described in detail.
7. Line 607: Incorrect figure reference.

Reviewer #2

(Remarks to the Author)

Reviewer #3

(Remarks to the Author)

The revised manuscript "Defining expansions and perturbations to the RNA polymerase III transcriptome and epitranscriptome by modified direct RNA nanopore sequencing" convincingly addresses all major and minor concerns raised in the original review, and I would like to thank the authors for their clear responses and the additional supplementary data provided.

This is a very interesting study that warrants publication in Nature Communications.

Reviewer #4

(Remarks to the Author)

Verstraten et al. present a novel direct RNA sequencing method for detection of Pol III-dependent transcripts. This allows identification of putative new Pol III-dependent RNAs and direct calling of pseudouridine and m6A modifications. The revised manuscript is significantly improved in regards to technical description and data presentation. New data includes La protein depletion and demonstrates this techniques utility in examining pre-tRNA stability. Overall I feel the revised manuscript is significantly improved and addresses all my prior concerns. My remaining concern is the discordant data regarding Pol II/III-dependency for different transcripts in Fig. 2B-C and Fig S4B-D. This does not require the authors perform additional experiments, rather clean up the data included and how it informs their interpretation of the novel transcripts. For instance, there are select transcripts which do not decrease in response to any of the inhibitor treatments (novel tx 5, 7) which is puzzling and could imply they are sequencing artifacts or have very long half-lives. Regardless it's hard to interpret their transcriptional dependency with the existing data. Additionally, the control (canonical Pol III) transcripts do not respond as expected to ML-60218 treatment. The authors may want to consider removing the ML-60218 data and instead using the alpha-amanitin dose concentration as the data is more convincing. Please address the following issues:

1. How do the authors explain the absence of a decrease in canonical Pol III-dependent transcripts (RN7SK, VTRNA, RMRP, 5S) after ML-60218 treatment. One option is simply that ML-60218 is not working as expected. Alternatively, the long half-lives of these transcripts may make it hard to observe changes in the current inhibitor treatment time line. In the future, consider assaying pre-tRNAs as a more responsive Pol III-dependent transcript control.
2. How do the authors explain novel tx 5 and 7 which have no significant decrease in any inhibitor treatment?
3. Why do select novel transcripts (tx 4, 3, 2) not have similar trends between Fig. 2C and Fig. S4D. Why did the authors change inhibitor concentration and time of addition between the experiments?

Version 2:

Reviewer comments:

Reviewer #1

(Remarks to the Author)

The authors have made additional progress in addressing our earlier concerns, most notably through the inclusion of three in vitro transcribed (IVT) tRNAs, which is essential given the study's emphasis on tRNA modifications. The overall manuscript is improved and the methodological contribution remains significant. We continue to view the selective sequencing of oligo(U)-tailed RNAs as a valuable advance with impact.

That said, my primary concern from earlier rounds remains largely unresolved and, if anything, has become more central: the justification of the 0.98 modification-probability threshold that defines what counts as a modification throughout the study. The authors' new "z-score of the difference" framework introduces a quantitative element to the selection of this cutoff, and I appreciate the effort to formalize it. However, as it is described in the methods section, this method functions mainly as a feature-scaling and visualization procedure, not as a statistical test or calibrated decision rule.

As described in the Methods:

"Statistical analysis of modification profiles: To assess differences in modification probability between experimental conditions and the IVT MIX control, we computed per-bin differences across the modification probability range. For each bin, the difference between the condition and IVT MIX was calculated and transformed into a z-score relative to the mean and standard deviation of all bin-wise differences. Bins with an absolute z-score > 2 were classified as significantly different from the control."

This provides a convenient way to visualize where biological and IVT datasets diverge (notably above 0.98), but it does not establish a null model or control a false-positive rate. Every downstream biological claim (the number and identity of modified sites, comparisons across conditions, etc.) rests on this threshold, which remains arbitrary and difficult to interpret. A more rigorous approach would use the IVT data as an empirical null to estimate the FPR across candidate cutoffs, selecting 0.98 based on a controlled error (rather than on internal "divergence" criteria).

Additionally, while the authors emphasize their ability to resolve tRNA modifications and now include three IVT tRNAs, the positional modification heatmaps for the individual tRNA types (endogenous versus IVT) are not included in the revision as they are for RN7SK and EBER2. Given the central focus on tRNAs, including these heatmaps are important for interpretation and should be included.

The authors also acknowledge that the La knockdown and other cell lines are represented by single replicates, this level of replication remains below what is generally expected in the genomics communities, and at a minimum should be explicitly acknowledged as a limitation for interpretation.

Minor points

Based on the figure legends, the y-axis in Fig. S5C appears to represent the "probability difference" rather than the "z-score of probability difference." This should be corrected or clarified in detail.

Reviewer #2

(Remarks to the Author)

Reviewer #5

(Remarks to the Author)

I was asked to advise on an opposing view between reviewer 1 and the authors. Let me start by stating that the quality of the reviews but also the quality of the efforts of the authors to respond to the comments are commendable.

The chief remaining concern of reviewer 1 is the motivation of the 0.98 threshold used as a modification-probability threshold to define what counts as a modification throughout the study. It is (rightfully) pointed out that i) this assumes proper calibration of Dorado's internal scoring and ii) only two RNAs with IVT data is available to demonstrate the validity of the threshold.

The authors have included an additional 3 pre-tRNAs IVT datasets to demonstrate broader validity of the 0.98 threshold. Furthermore, the authors have now included a z-score method to demonstrate that above the 0.98 threshold substantial z-scores are observed indicating where biological and IVT datasets diverge.

The reviewer (rightfully) points out that the z-score analysis does not replace a proper statistical framework that would guarantee a certain FPR.

It is my opinion that the z-score analysis indeed does not really add a lot, except enabling the visualization in Figure S5c, which is certainly intuitive and gives some credibility to the chosen 0.98 threshold. This z-score analysis should therefore not be overstated, which could lead to a false sense of 'statistical underpinning'.

The problem is that, unlike for DNA modification calling (5mC, 5hmC), there is no large scale 'ground truth' (usually synthetic

oligo-based) datasets available. This precludes proper statistical analysis of modification calling performance and analysis of potential context specificity or other biases. This is not something the authors can easily address. This limits the advance provided by the work somewhat.

Having said that, I find that sufficient circumstantial credibility is provided to support the conclusions. For instance, the divergence analysis demonstrates consistent results across all IVT datasets. Moreover, it should be noted that for DNA modification analysis Dorado's modified base probabilities are reasonably well calibrated.

Given the overall quality of the work and advance of the field, I would advise to include a proper discussion of the limitation and potential risk for biases on the downstream results in the discussion section.

The other remaining issues raised by reviewer 1 should be easily addressable.

We thank the editor and reviewers for the efforts in providing comments and critiques of our manuscript. We have included point-by-point responses below and would note that the manuscript has been extensively reconfigured with the introduction of additional figures and the reworking of many existing ones.

Reviewer #1 (Remarks to the Author):

The manuscript “Defining expansions and perturbations to the RNA polymerase III transcriptome and epitranscriptome by modified direct RNA nanopore sequencing” by Verstraten et al. implements a new method, which the authors define as DRAP3R (Direct read and analysis of Polymerase III transcribed RNAs), which essentially depletes non-poly(U) RNAs and ligates poly(U)-targeting adapters, thereby targeting Pol III-derived ncRNAs, which possess 3'-oligo(U) sequences owing to the oligoT termination mechanism of Pol III transcription. **The major strengths of the current study are (1) the direct sequencing of Pol III-derived RNA species, which provides some level of investigation regarding the chemical modifications of these oligo(U) RNAs - a major focus of the study - and (2) discovery of novel Pol III-derived RNAs.** The authors also use the DRAP3R framework to query changes in tRNA levels and pseudouridine stoichiometry following infection. The implementation of this method and the expansion beyond currently annotated Pol III-transcribed genes are valuable contributions that are certain to be of interest across the Pol III field and beyond. **However, there are notable concerns with the chemical modification (“epitranscriptome”) aspect of the current study that should be addressed, primarily with respect to the sparsity of in vitro transcribed, non-modified RNA species, which appear to be a critical requirement for interpreting all epitranscriptome results.**

We thank the reviewer for their comments and critiques which we address below. It is also clear that elements of our writing were not sufficiently clear and we have worked to improve this in the revised version.

For example, Figure 2C is our first introduction to inferred pseudouridine stoichiometry of in vitro transcribed 7SK RNA – it was surprising to see such a high level of pseudouridine stoichiometry for non-endogenous RNA, can the authors explain this?

The high level of pseudouridine stoichiometry in IVT 7SK (e.g. pos 91 and 92) is only observed in the unfiltered datasets and clearly demonstrates the very high false-positive rate associated with pseudouridine and m⁶A basecalling in nanopore DRS data. This is precisely why a filtering strategy to minimise false-positives, while retaining true-positives, is developed and applied here. When applying our filtering strategy (i.e. retaining only pseudouridine and m⁶A basecall confidence scores $\geq 98\%$), we observe the stoichiometry at all false positive positions called in the IVT datasets to be reduced to 0 (i.e. no pseudouridine). Crucially, these same filters, derived from our analysis in Fig 4a and Supplementary Fig. 5, enable the detection of pseudouridine at positions 243, 247, and 250 in all biological samples but not in the IVT sample. Importantly, this correlates with many previous studies of pseudouridine installation in RN7SK (PMID: 27558685 and 25192136).

While the manuscript notes that filtering reduces false positive rates, it suggests that in vitro transcribed RNA is needed for each and every RNA species in order to interpret modification state. This is also the case for m⁶A methylation, shown for 7SK in Supplemental Figure S4a, and for the only other RNA species with IVT - EBER2 (Supplemental Figure S4c,d).

To better explain how we used our IVT datasets to set the threshold for setting confidence scores in our biological datasets, we have explained this in more details in the revised

manuscript, lines 283 - 287. In addition, Figure 4a and S5b indicates that both IVT RNAs (EBER2 and 7SK) produce similar modification probability distributions to the biological samples for confidence scores below 0.98 at which points the biological samples and IVT samples diverge (i.e. in biological samples, there are relatively large numbers of modification calls with confidence scores > 0.98 that are not observed for the IVT datasets). This rationalises our choice of setting a confidence filter at 0.98. We certainly do not consider a requirement that every single RNA species requires an IVT RNA for comparison as the 7SK and EBER analyses both demonstrate that a confidence cutoff of 0.98 is sufficient to remove false-positives without compromising true-positive detection.

In addition to this apparent technical challenge, there is a general lack of statistical analysis, such that it remains difficult to interpret. For example, we are told that filtering of IVT false positives facilitates identification of modification state, but what statistical framework was applied to achieve this result? Naturally, these are critically important for interpreting the broader analyses focused on RNA modification - for example, are in vitro transcribed tRNAs not necessary for interpreting the results of Figure 3b (and supplemental Figure S6)?

We have now clarified the statistical rationale behind our filtering approach (lines 253-300). Specifically, we describe how we leverage the confidence scores provided by the native basecalling software (Dorado) to remove low-confidence calls and reduce false-positive rates i.e. we are building upon the statistical framework embedded within the basecaller. We have updated Fig. 4, S5, and S6 to better illustrate the effect of our filtering approach.

There are then a number of stoichiometry count and density analyses layered on top of this that are difficult to interpret. For example, on the basis of evidence provided for in vitro transcribed 7SK and EBER1, it is possible that analyses and interpretations have more to do with uridine or adenosine densities rather than any true endogenous modification states.

As above, we have worked to improve the clarity of the figures and associated text.

In the event that the technical challenges noted above cannot be addressed, I would reiterate that direct capture and sequencing of oligo(U) ncRNAs and discovery of novel (potentially) Pol III products is significant, and the authors might consider reworking their study to emphasize these findings more prominently.

We appreciate these positive comments and have made every effort to improve the clarity of the manuscript.

Major points:

1. There is a significant assumption that analyses are tied to nascent RNAs in the absence of any method to specifically isolate or enrich for nascent RNA. Though tRNAs undergo 3' cleavage, other Pol III-derived RNA species are not processed in the same manner, and thus oligo(U) alone is not sufficient to describe an RNA molecule as nascent. From the opposite perspective, the authors likely do not capture any mature or charged tRNA molecules, and thus all modification aspects related to the full dynamic pool of tRNAs is lost using this approach.

We appreciate this perspective which is similar to that provided by reviewer 3 and 4. On the recommendation of reviewer 4, we have altered the terminology, replacing 'nascent' with 'premature' throughout the manuscript. Regarding the absence of information from mature/charged tRNA molecules, we consider this a significant advantage of DRAP3R. We recommend that researchers interested in mature tRNAs instead use the nano-tRNA-Seq approach that was published last year (PMID: 37024678). By capturing premature tRNAs,

DRAP3R complements nano-tRNA-Seq by enabling the stratification of modifications added prior to or during 3' end processing and those which are added after – as demonstrated in Figure 5g in which we delineate between modifications present on pre-tRNAs and those which are present on mature tRNAs. Moreover, the reduced number of modifications present on pre-tRNAs reduces the signal complexity which would allow for better matches to the training data originally generated by ONT.

2. A statistical framework is necessary to interpret “highly confident modification sites”
As noted above, as part of the revision of the figures and text we have expanded upon the logic underlying our filtering approach and how this makes use of the statistical framework established in the native basecalling software.

3. There is no evidence provided in Figure 4 to support the claim that pre-tRNA levels are increasing while pseudouridine stoichiometries is decreasing upon HSV-1 infection. Projecting colors onto a correlation plot is not sufficient, instead a differential analysis is needed to draw “differential”-related conclusions. Though a volcano plot is provided in Supplemental Figure S7, we are not provided any details of what genes are up- and down-regulated.

The reviewer is of course absolutely correct on this matter. We have now added an additional biological replicate for HSV-1 infections harvested at 12hpi and have placed the relevant volcano plots in Fig 6c-d. We have also added supplementary tables containing the DeSeq2 outputs.

Minor points:

1. Figure 1b: The average read length for PolyA+ RNA seems to be abnormally short in length, can the authors explain?

We apologise for the lack of clarity. We had limited the y-axis scale to better show the distributions of read lengths derived from DRAP3R libraries. We have now included an alternative representation of the read length distribution for the poly(A) dataset in Supplementary Figure 1c.

2. Figure 1c, the legend color of the virus RNA does not appear to match.

We apologise for this error. In the revised version, viral reads (EBV EBERs) are shown in purple and, for Fig. 1, only appear in the CRO-AP5 #1 dataset.

3. Supplemental S1e is missing a legend.

We apologise for this oversight. We have reworked all legends in the manuscript as part of the revision.

4. Figure S2c, pol II-specific inhibitor can be used to verify that the novel transcripts are Pol III transcribed; the authors might also consider using ML-60218 for Pol III-specific inhibition

This is an excellent suggestion and we have now included an analysis of transcription for a selection of Pol II and Pol III targets in the presence of ML-60218. The results mirror those of the alpha-amanitin data presented previously and have been incorporated into our new Fig. 2 and Supplementary Fig. 4.

5. Figure 3a, Why is RNY4 listed separately from other Y RNAs? It might help to group together and label each Y RNA. In addition, specific RNA species are missing from this - perhaps most notable absent is 7SL (one of the most abundant Pol III transcripts). Can the authors comment on why 7SL was not captured (beyond the description in line 286)

We apologise for the confusion here. This figure only includes Pol III transcribed RNAs that contain at least one pseudouridine site with a stoichiometry above 10%. While 7SL is indeed captured by DRAP3R and therefore present in the main dataset, the premature RNA does not contain any pseudouridine sites > 10% stoichiometry and so is it is not included in Fig 5a. A second issue was that the Y RNA annotation in Fig 5a is incorrect and should in fact read RNY5. This has been corrected in the revised Fig 5a and line 340. Similar to 7SL, RNY1 and RNY3 premature RNAs do not contain any pseudouridine sites > 10% stoichiometry.

6. Figure 3b,c,f are missing legends (we are to assume it's the same as 3a).

We apologise for this oversight. We have reworked all legends in the manuscript as part of the revision.

7. Figure 4a. Given that samples are sequenced on individual minlons, how do the authors account for potential batch effects? This might be of particular concern if the libraries were sequenced at very different times, with potential for differences in minION batch and/or pore integrities. Overall, it is again difficult to “differential conclusions” drawn from single replicate comparisons, which the authors appear to be doing with respect to uninfected and infected samples.

In the absence of a fully supported multiplexing protocol for direct RNA sequencing, it is necessary to sequence each library on individual flowcells. While each flowcell differs in the numbers of pores available at the start of a run and this can impact on the total yield of sequencing reads, we observe no evidence that these differences would impact on the recorded composition of the libraries. Indeed, we include multiple biological replicates in this study (ARPE-19 #1 & 2, Figure 1; HSV-1 6hpi #1 & 2, HVS-1 12hpi #1&2, Fig. 6), all of which show very high correlation scores when comparing RNA abundances and pseudouridine stoichiometry distributions. We would also note there are many other published studies that have shown that any flowcell/pore-driven batch effects have no effective impact (e.g. PMID: 34850115, 38807060, 40082608)

Reviewer #2 (Remarks to the Author):

We thank the reviewer for efforts

Reviewer #3 (Remarks to the Author):

Verstraten et al. present DRAP3R, a nanopore direct RNA sequencing (DRS) methodology tailored for analyzing Pol III transcripts, leveraging their characteristic poly(U) tails. The

approach involves depletion steps, a custom adapter, and specific analysis strategies to identify transcripts and modifications like Ψ and m6A. The authors validate the method, identify novel transcripts, and apply it to study changes during HSV-1 infection. The data hint at some interesting changes to pol III transcript (relative) abundances and modification profiles upon HSV infection, but this would require the analysis of further biological samples to firm up conclusions.

Major Comments

1. Could the authors provide empirical evidence, perhaps via supplementary data, to demonstrate the specific advantages gained from the poly(A) and rRNA depletion steps, particularly given the use of a custom adapter designed for target enrichment? Quantifying the impact of these depletions on the yield and specificity of Pol III transcript capture would strengthen the methodological justification.

The goal of DRS library preparation is to maximise the amount of target (RNAs with adapters) within the total pool of RNA that is loaded onto a flowcell. This is particularly important because sequencing output decreases significantly if too much RNA is loaded into the flowcell. Thus, the aim of any DRS library preparation approach is to maximise the amount of target. For the standard DRS approach this is achieved by enriching for poly(A)+ RNAs prior to adapter ligation. In the absence of a functional enrichment approach for poly(U)+ RNAs, we designed a subtractive approach for DRAP3R. Omitting the poly(A)+ and rRNA-depletion steps would simply reduce the amount of target within the total pool of RNA loaded and would reduce the sequencing yield (number of reads).

2. The method's reliance on a 3' poly(U) tract means it may be biased against Pol III transcripts with non-canonical termini (e.g., resulting from imperfect termination). A discussion of this limitation, ideally supported by comparative data (e.g., using polyadenylation-based capture on the same polyA and rRNA depleted RNA), is warranted to assess the comprehensiveness of the captured transcriptome.

To the best of our knowledge, a key feature of RNA processing is the binding of protein La to the poly(U) tract present at the 3' end of all nascent Pol III transcribed RNAs. While the poly(U) tract may be degenerative (e.g. UCUU in the case of VTRNA1-3), the absence of such a tract altogether should render the precursor RNA unsuitable for processing. The presence of VTRNA1-3 at expected levels (relative to VTRNA1-1 and VTRNA1-2, PMID: 11479319) in our datasets confirms that our DRAP3R adaptor captures RNAs with degenerate 3' poly(U) tracts. We have highlighted this in lines 173-174.

3. Further justification is required for the chosen alignment parameters. Beyond increasing the percentage of aligned reads (Sup Fig 1c), how was the accuracy of these alignments confirmed, particularly considering the potential for spurious alignments suggested by the increase in secondary/supplementary hits?

Our criteria for identifying the optimal alignments parameters were defined by (i) the expectation that all sequenced reads should derive from the human genome, (ii) the alignment length should include be roughly 20-70 nt shorter than the read length. With this in mind it is clear that the bwa sw opt 2 and minimap2 approaches are not suitable due to the very low numbers of aligned reads. bwa sw opt1 results in many more alignments but a notable proportion of these are much shorter than the read lengths, indicative of spurious alignments, a feature that is also observed, albeit to a lesser extent for the bwa mem parameters. Importantly however, filtering to retain only primary alignments removes the

majority of these spurious alignments, significantly increasing our confidence in all primary alignments. This is now captured by comparisons of read vs alignment length shown for all examined parameters in the new version of Supplementary Fig S1 and is discussed in lines 140-144.

4. The rationale for selecting the 0.98 confidence threshold for modification calls (Ψ and m6A) needs clearer justification. An analysis demonstrating the sensitivity of key findings (like modification stoichiometry) to this specific cutoff value would strengthen the robustness of the conclusions.

We argue that the data presented in Fig. 4 and Supplementary Figs. 5-6 demonstrate exactly this. In the absence of a confidence threshold cutoff, numerous false-positive high-stoichiometry pseudouridine sites are detected in IVT 7SK and EBER2 RNAs. The application of filtering removes all of these sites. A similar observation is made for 7SK and EBER2 RNAs extracted from cells except that our filtering strategy still enables the detection of three well-established pseudouridine installation sites (243, 247, and 250).

5. The presence of intermediate-confidence m6A peaks is an interesting finding that merits further discussion. Do these signals exhibit unique basecalling error profiles, map to positions of other known modifications, or show enrichment in specific transcript types/classes?

We fully agree this is a fascinating observation. An additional analysis demonstrates that these peaks are only associated with read alignments that do not overlap with known Pol II or Pol III transcribed genes, nor any of the eight novel Pol III transcribed RNAs that we report in this study. This is now shown in Fig. 4b-c and discussed in the text on lines 263-271. Unfortunately, we are not currently able to isolate the specific reads/positions at which these occur due to inherent limitations of the ONT modkit software that is used to filter the data. We hope that newer versions of modkit will enable us to address this interesting question in the future.

6. Precision in terminology regarding modification timing and statements regarding sequencing of nascent transcripts. Is the evidence sufficient to distinguish installation 'during transcription and/or La binding' specifically. Is the protocol not capturing rather transcripts prior to 3' end processing?

We appreciate this perspective which is similar to that provided by reviewer 3 and 4. On the recommendation of reviewer 4, we have altered the terminology, replacing 'nascent' with 'premature' throughout the manuscript.

7. Statements regarding transcript expression changes upon HSV-1 infection require careful phrasing to reflect the relative nature of sequencing abundance data. Terms like 'reduces expression' or 'increases pre-tRNA levels' should be clarified as changes in 'relative abundance' within the sequenced library, acknowledging that absolute levels are not directly measured without appropriate normalization controls (which may be challenging here).

This is an excellent point and we have clarified the text as requested (line 386)

8. Conclusions drawn about time-dependent effects or correlations with infection length in the HSV-1 experiments should be presented cautiously, acknowledging the number of replicates and time points analyzed (6 hpi and 12 hpi). Explicitly stating these limitations would provide better context for the claims.

We have addressed these limitations in lines 409-411.

9. It may be worth explicitly noting in the limitations that, like standard DRS protocols, DRAP3R does not provide sequence information for the extreme 5' end of the RNA molecules.

We have added a figure panel to Supplementary Fig. S3b that shows this and included this limitation in lines 171-173.

10. Clarity of the figure legends should be improved throughout. Explicit definitions for all visual elements, particularly color coding (e.g., Sup Fig 1d lines, gene classifications in Fig 1c/e), are needed for unambiguous interpretation.

We apologise for this and have extensively rewritten figure legends to improve detail and clarity.

Minor Comments

1. Please reconcile the discrepancy between the adapter sequence described in the text (NNAAAA 3' overhang) and that depicted in Figure 1a (appears as AAAAAA).

This was an error in the graphic for Figure 1a and has now been corrected. We thank the reviewer for bringing this to our attention.

2. The read classification percentages presented in Figure 1c and discussed in the text (lines 162-189) could be presented more clearly to ensure all categories sum appropriately or unassigned portions are explicitly stated.

To address this point, we have included a new Supplementary Table S1 which contains the raw count data underlying Fig 1c and Fig 4b. We have also included heatmaps showing TPM values for all Pol III transcribed RNAs (primary Pol III genes and pre-tRNAs) across the different datasets as part of Supplementary Figures 3 and 10X.

3. In Sup Fig 1d, please clarify if the observed ~60-70nt difference between read length and alignment length corresponds primarily to the unaligned 3' adapter sequence, as expected.

Yes indeed, the reviewer is correct and we have now clarified this in lines 140-144

Overall Conclusion

This manuscript introduces a potentially valuable tool (DRAP3R) for studying the Pol III transcriptome and epitranscriptome using nanopore sequencing. Addressing the points raised above, particularly concerning methodological justifications, potential biases, bioinformatics parameter choices, and careful interpretation of the data, would significantly strengthen the study and increase confidence in its findings.

We greatly appreciate the reviewer's enthusiasm for the work and for the many helpful critiques. We hope they will be satisfied with how we have addressed them and will agree that the manuscript is now even stronger.

Reviewer #4 (Remarks to the Author):

Verstraten et al. present a novel direct RNA sequencing method for detection of Pol III-

dependent transcripts. The authors' approach is the first of its kind to assess this broad class of RNAs via direct RNA sequencing and uniquely facilitates identification of RNA modifications. The authors use their technique to assess three distinct human cell models in addition to IVT RNAs as a control to develop a tailored bioinformatics pipeline with high confidence dissection of these transcripts. This allows identification of putative new Pol III-dependent RNAs and direct calling of pseudouridine and m6A modifications. Their technique is corroborated by prior instances of RNA modification and provides clarity on discordant reports. Finally, they apply the technique to assess RNA shifts during viral infection and identify changes in RNA abundance and modification. The strength of this paper lies in development and benchmarking of the technique. The paper is well written and thorough in their representation of the data and how methodology was developed and verified.

We appreciate the reviewer's positive comments and their critiques which have enabled us to further strengthen the manuscript.

Major comments:

1. I would ask the authors to be more intentional with how they refer to the RNAs distinguished by their technique. I feel a more appropriate term would be premature rather than nascent. This technique successfully identifies RNAs which contain a polyU-track. For most Pol III-dependent transcripts, the 3' polyU tracks are removed during transcript maturation, however the exact rate at which they occur (as well as order relative to other maturation steps) is not clearly defined in humans. Or in some instances, such as U6, which the authors comment on in the discussion, polyU's are still present in the mature transcript.

We appreciate this thoughtful comment and agree that referring to the captured RNAs as 'premature RNAs' is more accurate than 'nascent RNA' and we have updated this throughout the manuscript.

2. In line with my prior comments, I feel the biological significance of this manuscript would be bolstered by perturbation of at least one of the core steps of Pol III transcript maturation (e.g. **La protein interaction**, nucleolytic cleavage) and **then assessment via DRAP3R**. Alternatively, the authors could apply their technique to different subcellular RNA fractions, as the location for some of these RNA modifications is unknown. Such experiments would significantly broaden the scope of this paper and inform the RNA biology of Pol III transcripts.

This is a great suggestion and we thank the reviewer for making it. To address this we reduced La protein levels by siRNA targeting and collected RNA at 72 hours post transfection. Knockdown was verified by Western Blotting and RNA was subjected to the DRAP3R protocol. Sequencing yielded almost one million reads that showed a similar distribution of read lengths to other DRAP3R datasets. We further observed a small reduction in the proportion of reads assigned to tRNA gene bodies and tested the hypothesis that, in the absence of La protein, pre-tRNA stability would be reduced. We calculated decay rates for each pre-tRNA and observed a significant increase in decay rates in the absence of La protein. Somewhat surprisingly, most other Pol III transcribed RNAs appear protected against degradation in the absence of La protein. We now hope to follow up on these findings with a more focused study aimed at understanding why some RNAs appear more resistant to degradation when La protein is depleted. This work is shown in Fig. 3 and described on lines 236-251.

3. The authors present their dataset in a number of interesting ways, I would ask that they expand on the following topics:

a. Short RNA reads (line 186-187): The authors remark on a proportion of reads which are quite short. I'm curious what these reads map to (type of transcript, location in transcript, etc)

and if the stop location may indicate an RNA modification that their reverse transcriptase could not proceed through. Many other studies of Pol III dependent or other highly structured RNAs use a TGIRT-based RT step to aid with processivity. Please expand on these short products, including their parent gene loci, size, and whether they're stop position correlates with previously published RNA modifications.

The reviewer is referring to our intersect analysis in which aligned reads are counted only if their alignment covers at least 25% (0.25) of the annotated gene. For the primary Pol III genes and tRNA genes, only a few thousand reads (less than 0.1 - 0.5% of all Pol III derived RNAs) did not satisfy this criterion. The distribution of discarded alignments did not differ from those retained. By contrast, applying these parameters to Pol II derived transcripts resulted in ~60% of alignments to Pol II genes being categorised as 'too short'. This makes good sense given that only Pol II derived RNAs that are poly(U) tailed (templated or non-templated) would be captured and many of these (particularly those targeted by polyuridylation) are likely 5' degraded. Finally, we agree with the reviewer that the use of a TGIRT-based RT makes good sense and is something we plan to test for DRAP3R in the future, especially since ONT have also switched to a TGIRT-based RT for their DRS protocol (NEB Induro Reverse Transcriptase). However, the evidence to date suggests that we do not observe issues with processivity during the DRAP3R library preparation.

b. Pol II RNAs (line 183-185): Polyuridylation is a form of RNA processing by which U's are added to the 3' end of mRNA or ncRNA. This would be something I'd expect the authors could identify in their datasets for Pol II products by comparing the gene sequence to that of the potential 3' overhang on their direct RNA read. This would be another very interesting benefit of their novel technique. Please include an analysis or discussion of whether their technique can detect this 3' RNA processing event.

This is a great comment. We do indeed observe that DRAP3R captures specific mRNAs and ncRNAs that have previously been identified as targets for polyuridylation including HIST2H3D/H3C13 (PMID: 18172165) and pre-let-7 (PMID: 35797480). We have included this on lines 191-194 and also highlight that DRAP3R might be adapted for polyuridylation studies in the discussion (lines 466-468)

c. Viral RNAs (Fig 4b): It's a bit hard to see with the color scheme of the column bar, but it appears the authors identified reads mapping to the virus in their dataset—at least there's a legend box for virus. Please expand on these reads and whether they present as bonafide Pol III-dependent transcripts

We have expanded on this data by including coverage plots across the HSV-1 genome for each of the samples included. While prior ChIP-Seq studies have indicated that Pol III can bind the HSV-1 genome (PMID: 35110532), the transcription patterns observed are not consistent with those observed for other DNA viruses with Pol III genes (e.g. EBV EBER). An unusual feature of the HSV-1 genome is the presence of numerous short homopolymer repeats (PMID: 24227835) which, in the case of thymine, would provide templates for the DRAP3R adapter. We would thus postulate that the reads derived from HSV-1 are in fact Pol II transcribed RNAs captured mid-transcription of poly(T) homopolymers. This is now discussed in lines 412-420.

Minor comments:

1. We would ask the authors to comment on how quantitative DRAP3R is. It does not appear that they introduced IVT spike-ins at early steps in the protocol, thus I am apprehensive of

abundance comparisons in an experimental condition such as infection would robustly changes the RNA landscape. This does not change the major conclusions of this paper, but rather presents as a current study limitation that the authors should comment on.

This is a good point and something we hope to develop for DRAP3R at a later stage. We have added this limitation to the discussion (lines 468-471).

2. To support the authors findings regarding novel Pol III transcripts (Fig. S2), I would ask that they integrate mapping of published ChIP-Seq datasets for the Pol III and II machinery at these gene loci. This would support conclusions regarding RNA polymerase dependency and suggest a classification (Type 1, 2, 3) for Pol III RNAs.

In principle this is an excellent idea. However, there remains a lack of a functional repository containing the relevant information (i.e. locations at which different elements of Pol III and Pol II machinery bind to the genome). We did attempt this analysis using Pol3base but this database is also not functional. However, of the eight novel transcripts identified here, three are implicitly supported by previous studies (novel tx 1, 2, and 4). To get at this a different way, we followed the suggestion of reviewer 1 and used a Pol III inhibitor to determine which novel transcripts are sensitive to inhibition. Consistent with the results from alpha-amanitin treatment, we observed that at least four of the novel tx are sensitive to Pol III inhibition. This is shown in the new Supplementary Figure S4. We have also moved the original supplementary Figure S2 into the main text as part of this revision and it is now listed as Fig. 2.

3. Fig. 1C and 4B. Please add a heatmap to expand the data presented in the column bar graphs regarding the RNAs represented in the Pol III-dependent categories.

This has now been added to Supplementary Figures S3 and S10

4. Fig. 1C legend: Can the authors please define what they classify as “Pol III primary” or “Pol III pseudogene”, I’m assuming they refer to 5S rRNA, but the existing reference is ambiguous.

Our categorisation of Pol III primary and Pol III pseudogene is based on the classification provided in the gencode v45 annotation file (https://www.gencodegenes.org/human/release_45.html). Here, pseudogenes incorporate P into the name enabling the distinction of, for instance, RN7SL1 and RN7SL12P.

5. Please comment on the potential mechanism by which the virus alters Pol III-dependent RNA modification status. For example, what is known about the relative abundance or localization of the modifying enzymes.

This is an excellent question. Based on our previous studies of m6A regulation during HSV-1 infection, we would hypothesize that, similarly to the m6A methyltransferase subunits METTL3 and METTL14 (PMID: 34282019), pseudouridine synthases are relocalised to the cytosol as a means to restrict the installation of modifications. We are now testing this hypothesis as part of a follow-up study.

We thank the editor and reviewers for the efforts in highlighting critiques of our manuscript that were not fully resolved. We have addressed all of these and include point-by-point responses below.

REVIEWER COMMENTS

Reviewer #1 (Remarks to the Author):

The strength of the DRAP3R method reported by Verstraten et al. lies in its targeted capture and sequencing of poly(U)-tailed RNAs. Though the revised manuscript submitted by the authors makes some improvement in clarity, my primary concern remains with the interpretation of the epitranscriptome data, particularly for tRNAs – which was not addressed. In short, the authors set an empirical modification probability threshold (≥ 0.98) on the basis of rudimentary observational comparisons of endogenous and IVT 7SK, where no modifications should be expected. At this threshold, many putative modification sites in the endogenous data “disappear” in the IVT data, leaving three modification sites in 7SK that are also documented in the literature. This is presented as a “positive control” supporting the threshold choice, but it is shown only qualitatively, via a heatmap, without statistical or quantitative description of enrichment over IVT.

The “0.98” threshold is not part of a formal statistical framework applied to the data, but rather an empirical filter imposed on top of the Dorado basecaller’s internal scoring. Dorado’s modification confidence scores are posterior probabilities as estimated by the model for a given base in a given sequence context, and they are inherently model- and context-dependent, not universally calibrated across RNA types. In their rebuttal, the authors describe their filter as “building upon the statistical framework embedded within the basecaller,” but this conflates the basecaller’s internal probability model with their own threshold choice. The cutoff of 0.98 is derived from two RNAs (7SK and EBER2) where IVT data are available, based on visual divergence in score distributions, without quantitative estimates of false-positive reduction or true-positive retention.

Our analysis of basecall (inc. modifications) probabilities demonstrated that canonical nucleotides (ACGU) are generally called with confidence scores exceeding 95% (Fig. S5a). However, when examining pseudouridine and m6A calls, the confidence score distributions were very different with only a fraction of the modification calls showing confidence scores exceeding 98% (Fig. S5b). This observation led us to generating two IVT RNAs to determine whether lesser (i.e. <98%) confidence scores might reflect false positives (Fig. S5b). Here, we observed that pseudouridine modification probability distributions were near-identical between our biological samples and IVT samples for all values below ~ 0.98 but diverged significantly after that (Fig. S5b and Fig. 4a). Using RN7SK and EBER2 as examples, we observed a high rate of false positive pseudouridine calls in the IVT datasets that are mirrored across all our biological datasets. Application of the empirically derived filter (0.98) was sufficient to remove these false-positive sites from both RNAs without compromising detection of previously identified pseudouridine sites. Thus, it is important to emphasise that the 0.98 filter was not derived by looking for a value that only retained known pseudouridine sites but rather that our filter was derived from looking at deviations in the confidence score distributions. The fact that this produced a positive result for the EBER2 and RN7SK data provides further support for our choice of filter.

However, the reviewers concern about the empirical nature of the filter value is warranted and we have addressed this as follows. Firstly, we generated a replacement IVT dataset consisting of RN7SK, EBER2, and three pre-tRNAs (tRNA-Arg-ACG-1-1, tRNA-Glu-TTC-2-1,

and tRNA-Glu-CTC-1-1) which were sequenced as a single pool (Figure S1b). We subsequently generated new modification probability distribution plots that incorporate this dataset (Figure 4a and the new panel Fig S5c). We then implemented a quantitative divergence analysis for comparing the biological samples against the pooled IVT control. Here, we computed per-bin differences and converted these to z-scores, flagging bins where $|z| > 2$ as divergent (equivalent to selecting the top 5% of data). This approach provides a statistical and reproducible framework to identify modification probability values that diverge from the (IVT) background. Visualization of z-scores is now included in Figure S5c and revealed 0.98 to be an appropriate filter for identifying meaningful differences between biological and pooled IVT datasets.

While 0.98 may separate modified from unmodified at a handful of documented sites in 7SK and EBER2, there is no evidence this value is optimal or valid for tRNAs, which differ substantially in sequence, structure, and modification density. The same threshold is then applied to tRNAs without tRNA-specific IVT controls or orthogonal validation. The resulting tRNA profiles and conclusions are therefore based on an untested extrapolation, which was not truly quantitative to begin with.

As outlined above, we now have data from three IVT tRNAs that have been analysed with and without filtering. The results demonstrate a high false positive rate for pseudouridine detection in unfiltered datasets which is almost entirely eliminated using the 0.98 filter derived from our quantitative divergence analysis. This is captured in a new figure panel (Figure S8e). Importantly, our analysis of five distinct IVTs have consistently demonstrated (i) a high false positive rate in unfiltered data and (ii) a single threshold for filtering is sufficient to remove/minimise false positives without compromising true positive detection.

A further major limitation is the lack of biological replicates for many of the modification analyses. The threshold choice, the distribution comparisons between endogenous and IVT RNAs, the tRNA modification profiles, and the infection versus mock modification differences are largely drawn from single-sample datasets or comparisons. Without replication, it is not possible to distinguish genuine biological differences from run-to-run variation in nanopore signal, capture efficiency, or basecalling. This limitation restricts interpretation of differences in modification stoichiometry between conditions or transcript classes.

While we appreciate the reviewers concern regarding biological replicates, we would point out that we have biological duplicates included for multiple conditions. This includes (i) two biological replicates of uninfected ARPE-19 cells which show excellent correlations in terms of RNAs identified and their relative abundances (Fig. 1b, c, and e), as well as two biological replicates each of HSV-1 infected ARPE-19 cells collected at 6hpi and 12hpi (Fig. 7a and b, Fig S10a). Thus, the changes in tRNAs abundances and pseudouridine modification profiles described in HSV-1 infection are supported by biological replicates. Similarly, while most of the data shown for tRNAs in Fig. 5 is from one of the ARPE-19 datasets, the results are consistent with its biological replicate as well as the datasets generated from other cell lines (Fig. 5a, Fig. S8a, Fig. S9). Thus, from the three distinct sets of biological replicates generated as part of this study, we observe only very minimal variation.

Experimentally, the authors have now introduced La knockdown experiments (single replicate) which leads to additional, potentially overdrawn conclusions – in this case about decay rates. However, the authors do not comment on the apparent shift in read length distributions in the La knockdown data, which warrants careful consideration. The knockdown condition appears enriched for shorter reads compared to controls, which could directly influence metrics such as calculated decay rates when these are derived from early-to-late

gene read ratios. Without controlling for read length distribution, it is unclear whether the reported changes in decay reflect true biological effects or are instead a consequence of altered fragment size profiles.

The reviewer notes an apparent shift in the read length distribution and what this would mean for the decay rate calculations. We understand this is derived from an observed increase in raw read lengths around 70nt shown in Fig. 1b of the manuscript. However, as these raw read length distributions includes the adapter sequence (~70nt in length), we can surmise this peak indicates an increase in adapter-only reads generated during the nanopore sequencing. A better metric for judging differences in fragment size profiles is to look at alignment lengths (see Fig. R1 below). Here we observe very minimal differences in alignment length distributions between the control and siLa knockdown datasets. In addition, the model used to calculate decay rates calculates relative decay of coverage across the transcript, a procedure that would be insensitive to smaller differences in read length distributions.

Fig. R1. Alignment length distributions for select ARPE-19 datasets.

We remain convinced that the study is valuable for (1) its methodological advance in purifying and sequencing poly(U)-tailed RNAs, (2) the discovery of novel Pol III transcribed RNA species, and (3) its potential for predicting candidate modification sites from direct RNA nanopore data.

However, several descriptive conclusions remain insufficiently supported by the current data: (1) tRNA modification profiles are presented without tRNA-specific control experiments, (2) conclusions from certain perturbations, such as La knockdown, are based on non-replicated datasets, and (3) potential confounding effects of non-uniform read length distributions are now introduced and not addressed.

We appreciate the reviewer's positive assessment of the study as a whole. Regarding the three concerns summarised above we have now (1) included a set of IVT pre-tRNAs in the analysis which provide further support for our filtering strategy, (2) implemented a quantitative approach to defining optimal filtering values, (3) highlighted that our key experiments with uninfected and HSV-1 infected ARPE-19s are in fact supported by biological replicates, and (4) demonstrated that the aligned read length distributions are in fact near-uniform and thus would not impact decay rate calculations.

Minor comments

1. Replication: The APE-19 siLA and NHDF CRO-AP5 datasets each appear to have only a single replicate.

This is correct. We have two biological replicates each for the HSV-1 infected (2x 6hpi and 2x 12hpi) and uninfected ARPE-19 datasets, and these consistently showed excellent correlations in terms of the identity and abundances of RNAs detected.

2. Figure 3d (decay rate): Some decay rate values are shown as less than zero; clarification is needed here and in the methods.

The reviewer's point is well taken. Indeed, all valid decay rates should have values > 0 . The range of negative values initially obtained were due to an error in our decay calculation script in which we calculated changes in coverage in the 5' \rightarrow 3' direction rather than the other way around. This led to negative k values because coverage was increasing. We corrected this error, increased the minimum coverage to 100, reanalyzed the data, and updated Fig. 3 and the methods (line 636). The results remain the same except that all k values are now positive.

3. Figure 3e–h: The figure legend does not specify the meaning of the different colors or indicate the directionality of the tRNAs.

We apologise for this oversight and have updated the figure panels and legend to address this

4. Lines 286–287: Incorrect figure reference.

Thank you for flagging this, the reference has now been corrected

5. Line 395: Incorrect figure reference.

Thank you for flagging this, the reference has now been corrected

6. Figure 4e / Lines 325–326: The cutoff used for stoichiometry to designate a modification site is unclear. While a threshold is stated at Line 324, the rationale should be described in detail.

We set a 10% stoichiometry threshold because prior nanopore and targeted-quantification studies recover 10% synthetic mixtures reliably and BID-seq/BACS-style approaches treat sites above $\sim 10\%$ as confident, reproducible modifications (PMIDs: 36302989, 39349603, 39406497). This is now explicitly stated in the manuscript on lines 326-328.

7. Line 607: Incorrect figure reference.

Thank you for flagging this, the reference has now been corrected

Reviewer #2 (Remarks to the Author):

We thank the reviewer for their efforts

Reviewer #3 (Remarks to the Author):

The revised manuscript "Defining expansions and perturbations to the RNA polymerase III transcriptome and epitranscriptome by modified direct RNA nanopore sequencing" convincingly addresses all major and minor concerns raised in the original review, and I would like to thank the authors for their clear responses and the additional supplementary data provided.

This is a very interesting study that warrants publication in Nature Communications.

We appreciate the reviewer's enthusiasm for our work and thank them for their efforts.

Relating back to the comments provided in the first round of review, we were asked to provide more information on the double peak structure observed at low modification probabilities in m6A datasets (Fig 4a). In the first revision we demonstrated that this was arising only from reads aligning to unannotated regions of the genome. However, we also observe this feature in our pooled IVT dataset when aligning these reads against the genome (but not when aligning against a limited transcriptome containing just the five target sequences). Together this suggests these peaks are an artefact rather than being indicative of other adenosine modifications. We have included this analysis in the latest version of the manuscript (Fig 4c) and updated lines 272-275 in the text.

Reviewer #4 (Remarks to the Author):

Verstraten et al. present a novel direct RNA sequencing method for detection of Pol III-dependent transcripts. This allows identification of putative new Pol III-dependent RNAs and direct calling of pseudouridine and m6A modifications. The revised manuscript is significantly improved in regards to technical description and data presentation. New data includes La protein depletion and demonstrates this techniques utility in examining pre-tRNA stability. Overall I feel the revised manuscript is significantly improved and addresses all my prior concerns.

We greatly appreciate the reviewer's enthusiasm regarding the revised version of the manuscript.

My remaining concern is the discordant data regarding Pol II/III-dependency for different transcripts in Fig. 2B-C and Fig S4B-D. This does not require the authors perform additional experiments, rather clean up the data included and how it informs their interpretation of the novel transcripts. For instance, there are select transcripts which do not decrease in response to any of the inhibitor treatments (novel tx 5, 7) which is puzzling and could imply they are sequencing artifacts or have very long half-lives. Regardless it's hard to interpret their transcriptional dependency with the existing data. Additionally, the control (canonical Pol III) transcripts do not respond as expected to ML-60218 treatment. The authors may want to consider removing the ML-60218 data and instead using the alpha-amanitin dose concentration as the data is more convincing. Please address the following issues:

1. How do the authors explain the absence of a decrease in canonical Pol III-dependent transcripts (RN7SK, VTRNA, RMRP, 5S) after ML-60218 treatment. One option is simply that ML-60218 is not working as expected. Alternatively, the long half-lives of these transcripts may make it hard to observe changes in the current inhibitor treatment time line. In the future, consider assaying pre-tRNAs as a more responsive Pol III-dependent transcript control.

We agree with reviewer in regard to their concerns over the effectiveness of the ML-60128 inhibitor which, in our hands, does not consistently downregulate expression of the Pol III genes profiled. Assaying pre-tRNAs is an excellent suggestion and one we will incorporate in our ongoing optimisation efforts. Here, we agree that the best course of action is to remove the ML-60218 data from the current revision and to focus on the alpha-amanitin data. Fig. 2C and Fig. S4D have now been removed and the manuscript updated to reflect this.

2. How do the authors explain novel tx 5 and 7 which have no significant decrease in any inhibitor treatment?

Referring back to point 1, we have now removed the ML-60218 data.

3. Why do select novel transcripts (tx 4, 3, 2) not have similar trends between Fig. 2C and Fig.

S4D. Why did the authors change inhibitor concentration and time of addition between the experiments?

Referring back to point 1, we have now removed the ML-60218 data.

Dear Professor Depledge,

Thank you for submitting your manuscript "Quantitative profiling of RNA modifications in an expanded RNA polymerase III transcriptome" to Nature Communications. I am delighted to say that we are happy, in principle, to publish it under an open access license.

First, we ask you to revise your paper to address our editorial requests (in the attached Author Checklist) and any remaining comments from reviewers (included at the end of this email, if applicable). Please include a discussion of the limitation of the threshold approach and the potential risk for biases on the downstream results in the Discussion section. Please include heatmaps for tRNAs in the paper. Please mention limitation of single replicate in La knockdown experiments.

We thank the editor for their support and are delighted about the manuscript being accepted (in principle). We have now included:

- 1: The requested discussion of the limitation of the threshold approach and the potential risk for biases on the downstream results (lines 459 – 468)
- 2: Mention of the limitation of single replicates in the La knockdown experiments (lines 235-237).
- 3: The requested heatmaps for the tRNAs were already included in the previous revision (Supplementary Figure S8) but this was perhaps missed by the reviewer #1.

REVIEWERS' COMMENTS

Reviewer #1 (Remarks to the Author):

The authors have made additional progress in addressing our earlier concerns, most notably through the inclusion of three in vitro transcribed (IVT) tRNAs, which is essential given the study's emphasis on tRNA modifications. The overall manuscript is improved and the methodological contribution remains significant. We continue to view the selective sequencing of oligo(U)-tailed RNAs as a valuable advance with impact.

That said, my primary concern from earlier rounds remains largely unresolved and, if anything, has become more central: the justification of the 0.98 modification-probability threshold that defines what counts as a modification throughout the study. The authors' new "z-score of the difference" framework introduces a quantitative element to the selection of this cutoff, and I appreciate the effort to formalize it. However, as it is described in the methods section, this method functions mainly as a feature-scaling and visualization procedure, not as a statistical test or calibrated decision rule.

As described in the Methods:

"Statistical analysis of modification profiles: To assess differences in modification probability between experimental conditions and the IVT MIX control, we computed per-bin differences across the modification probability range. For each bin, the difference between the condition and IVT MIX was calculated and transformed into a z-score relative to the mean and standard deviation of all bin-wise differences. Bins with an absolute z-score > 2 were classified as significantly different from the control."

This provides a convenient way to visualize where biological and IVT datasets diverge (notably above 0.98), but it does not establish a null model or control a false-positive rate. Every downstream biological claim (the number and identity of modified sites, comparisons across conditions, etc.) rests on this threshold, which remains arbitrary and difficult to interpret. A more rigorous approach would use the IVT data as an empirical null to estimate the FPR across candidate cutoffs, selecting 0.98 based on a controlled error (rather than on internal "divergence" criteria).

Additionally, while the authors emphasize their ability to resolve tRNA modifications and now include three IVT tRNAs, the positional modification heatmaps for the individual tRNA types (endogenous versus IVT) are not included in the revision as they are for RN7SK and EBER2. Given the central focus on tRNAs, including these heatmaps are important for interpretation and should be included.

The authors also acknowledge that the La knockdown and other cell lines are represented by single replicates, this level of replication remains below what is generally expected in the genomics communities, and at a minimum should be explicitly acknowledged as a limitation for interpretation.

We thank the reviewer for their considered response. It is clear that the lack of large ground truth datasets remains an unsolved problem in the area of RNA modification detection and that all existing approaches, including ours, remain imperfect. However, as stated by reviewer #5, our divergence-based analysis does broadly justify our choice of probability cut-off for filtering modification calls. We have addressed this further in the discussion, highlighting the need for caution in interpreting results and the need for further advances in the field (reliable orthologous approaches and ground truth datasets). We further address the limitations associated with single replicates. Regarding the request for heatmaps showing the effect of filtering on IVT tRNAs and their biological counterparts, this was already included in Figure S8e of the previous revision.

Minor points

Based on the figure legends, the y-axis in Fig. S5C appears to represent the "probability difference" rather than the "z-score of probability difference." This should be corrected or clarified in detail.

We consider z-score of divergence to be the most accurate description as the axis is showing z-scores that are derived from a comparison of distributions. We have updated Fig. S5C to reflect this.

Reviewer #2 (Remarks to the Author):

Reviewer #5 (Remarks to the Author):

I was asked to advise on an opposing view between reviewer 1 and the authors. Let me start by stating that the quality of the reviews but also the quality of the efforts of the authors to respond to the comments are commendable.

The chief remaining concern of reviewer 1 is the motivation of the 0.98 threshold used as a modification-probability threshold to define what counts as a modification throughout the study. It is (rightfully) pointed out that i) this assumes proper calibration of Dorado's internal scoring and ii) only two RNAs with IVT data is available to demonstrate the validity of the threshold.

The authors have included an additional 3 pre-tRNAs IVT datasets to demonstrate broader validity of the 0.98 threshold. Furthermore, the authors have now included a z-score method to demonstrate that above the 0.98 threshold substantial z-scores are observed indicating where biological and IVT datasets diverge.

The reviewer (rightfully) points out that the z-score analysis does not replace a proper statistical framework that would guarantee a certain FPR.

It is my opinion that the z-score analysis indeed does not really add a lot, except enabling the visualization in Figure S5c, which is certainly intuitive and gives some credibility to the chosen 0.98 threshold. This z-score analysis should therefore not be overstated, which could lead to a false sense of 'statistical underpinning'.

The problem is that, unlike for DNA modification calling (5mC, 5hmC), there is no large scale 'ground truth' (usually synthetic oligo-based) datasets available. This precludes proper statistical analysis of modification calling performance and analysis of potential context specificity or other biases. This is not something the authors can easily address. This limits the advance provided by the work somewhat.

Having said that, I find that sufficient circumstantial credibility is provided to support the conclusions. For instance, the divergence analysis demonstrates consistent results across all IVT datasets. Moreover, it should be noted that for DNA modification analysis Dorado's modified base probabilities are reasonably well calibrated.

Given the overall quality of the work and advance of the field, I would advise to include a proper discussion of the limitation and potential risk for biases on the downstream results in the discussion section.

The other remaining issues raised by reviewer 1 should be easily addressable.

We thank the reviewer for their considered response. We agree that the continued lack of 'ground truth' datasets for RNA modification benchmarking remains a general problem faced by the field. We are also in agreement that explicitly stating potential caveats and risks of bias in the discussion is an important addition and we have now included this.